# Consumption responses to an unconditional child allowance in the United States

**Zachary Parolin** [1] ✉, **Giulia Giupponi** [1], **Emma K. Lee**[2] **& Sophie Collyer**[3]

The COVID-19 pandemic put families in the United States under financial stress. The federal government's largest response in 2021 was the American Rescue Plan Act, which temporarily expanded the Child Tax Credit (CTC) into a large, unconditional child allowance providing monthly payments to families with children. This study investigates consumption responses to the CTC expansion using anonymized mobile-location data and debit/credit card data that track visits and spending at 1.3 million establishments across US counties. For identification, we exploit variation in the size of households' income gains due to the CTC across counties in a difference-in-differences framework spanning January 2021 to May 2022. Counties benefiting most from the CTC expansion experienced larger increases in visits to childcare centres and health- and personal-care establishments, and increased visits to and spending per transaction at grocery and general stores. These findings suggest that the CTC expansion increased household consumption and spending on children.

Most high-income countries offer a regular cash benefit to families with children that is not conditional on employment[1]. Studies have shown that these 'child allowances' generally contribute to lower rates of poverty and hardship among families with children[2]. The United States is unique among its peer countries in not offering an unconditional child allowance at the national level; instead, its primary income supports for families with children are either conditional on meeting an earnings requirement (for example, Earned Income Tax Credit (EITC)), conditional on work-participation activities and narrowly targeted at single parents (for example, Temporary Assistance for Needy Families), or are provided as in-kind benefits (for example, Supplemental Nutrition Assistance Program)[3,4]. In March 2021, however, the federal government passed the American Rescue Plan Act (ARP), which transformed the Child Tax Credit (CTC)—normally a partially refundable, lump-sum tax credit for families with children—into a fully refundable child allowance[5,6]. The Internal Revenue Service (IRS) distributed payments to more than 60 million children (around 90% of US children) in two forms: half of the benefits were distributed in monthly instalments of up to $300 per child per month from July to December 2021, whereas the other half was distributed in a single lump-sum payment upon filing taxes[7].

Studies have shown that this temporary, unconditional child allowance was effective at reducing poverty and food hardship among low-income families with children[8,9]. Less is known, however, about families' specific consumption responses to the child allowance. Available nationally representative evidence on consumption responses comes from the Census Household Pulse Survey (Pulse) and Consumer Expenditure Survey (CES), but with strong limitations: both are survey based, which rely on respondents properly recalling and reporting their expenditure patterns, often aggregated to broad categories, and with the threat of recall error and/or social desirability bias[10]. The latter concern is especially stark for sensitive consumption categories, such as alcohol or tobacco expenditures[11]. Moreover, recent studies have shown that surveys on individual behaviours during the era of coronavirus disease 2019 (COVID-19) often suffer from low response rates or other threats to representativeness that challenge the validity of survey findings[12,13].

This study investigates consumption responses to the 2021 CTC expansion while avoiding most of the concerns that affect survey-based consumption reports. We use anonymized mobile-location data and debit/credit card data to track visits to and spending at 1.3 million establishments across counties that cover virtually all the US population.

[1]Department of Social and Political Sciences, Bocconi University, Milan, Italy. [2]Opportunity Insights, Harvard University, Cambridge, MA, USA. [3]Center on Poverty and Social Policy, Columbia University, New York, NY, USA. ✉e-mail: Zachary.parolin@unibocconi.it

For identification, we exploit variation in treatment intensity across counties stemming from differences in the income gains after the switch from the regular to the expanded CTC. Applying a series of difference-in-differences estimates, we study (1) the child allowance's impact on in-person visits—a proxy for consumption at the extensive margin—to seven establishment types, ranging from childcare centres to alcohol, tobacco and gambling venues; and (2) its impact on consumption at the intensive margin, measured as mean spending per transaction. We assess the robustness of our findings across a broad range of sensitivity tests and alternative specifications.

We show that counties benefiting the most from the CTC expansion experienced larger increases in visits to childcare centres and personal- and health-care establishments, and increased visits to and spending per transaction at grocery and general stores.

## Results

### Background

We provide technical details on ARP's policy changes to the CTC in Supplementary Appendix A; here we emphasize the three most important alterations. First, the ARP made the CTC fully refundable and ended the earnings requirement, meaning that the benefit was no longer conditional on earned income. Before the ARP, 1 in 3 children under age 17 did not receive the full CTC benefit value because their families did not earn enough to qualify, and children aged 17 and over were entirely ineligible for the credit[14,15]. The ARP extended the full credit to all low-income families with age-eligible children (which, in the expansion, also included 17 year olds). Second, the ARP increased the maximum benefit levels from $2,000 per year per qualifying child to $3,600 per year for a child under 6, and to $3,000 for children aged 6–17 (ref. 5). Third, the ARP changed the benefit distribution schedule: half the CTC benefit was provided in monthly instalments between July and December 2021, whereas the other half was provided in a single lump-sum payment when families filed taxes in the spring of 2022.

Strong regional heterogeneity exists in the share of tax units with children left out of receiving the full CTC before ARP[16]. The expanded CTC thus benefits a greater proportion of families with children in some counties than others. As we document in Supplementary Appendix B, the mean county-level gain in family income due to the CTC expansion is very strongly correlated with the county-level poverty rate. In tax units with children, the annual dollar value of added benefits due to the CTC is $2,472 in counties in the bottom tercile of the poverty rate distribution and $3,013 in the top tercile. The corresponding relative gains in income due to the expanded CTC are 4.7% and 8.7%, respectively. Although county-level income gains from the CTC are not estimable for all counties using public data, the US Census Bureau provides poverty estimates for the full list of US counties, estimated from American Community Survey (ACS) data. Hence, to maximize the geographic coverage of our analysis, we use the county-level poverty status as our proxy for the monetary and proportional gains from the expanded CTC. In Supplementary Appendix B, we show that this proxy strongly predicts the magnitude of relative and absolute CTC gains among the counties in our sample. At the same time, we show that the county poverty rate is not systematically associated with differential take-up of the expanded CTC.

We measure consumption responses to the temporary child allowance in two ways. First, we measure seasonally adjusted in-person visits to specific establishment types in each county and month, providing a measure of implied consumption on the extensive margin. Table 1 provides a list of the establishment types in our study. For example, we can measure the number of in-person visits to formal childcare centres in each county and each month between January 2019 and May 2022. Second, we measure spending amounts (via debit or credit card) at (a subset of) the same set of establishment types between January 2021 and May 2022, providing a measure of direct consumption volumes on the intensive margin.

**Table 1 | Groupings of establishment types and sample sizes**

| | Category | Examples (sample size, $n$) |
|---|---|---|
| 1 | Cars, clothes, hobbies and home | Car dealers, automobile parts and tyre stores; clothing stores and shoe stores; book stores, sporting goods stores, and toy and games stores; furniture stores, electronics and home appliances stores, lawn and garden equipment, hardware stores, and paint and wallpaper stores ($n$=216,889) |
| 2 | Grocery and general stores | Supermarkets, convenience stores, fruit and vegetable markets, and baked goods stores; department stores and general merchandise stores ($n$=170,168) |
| 3 | Alcohol, tobacco and gambling | Beer, wine and liquor stores; tobacco shops; casinos and casino hotels ($n$=72,641) |
| 4 | Health and personal care | Doctors' offices, dentists' offices and family planning services; pharmacies, cosmetics stores, other health and personal-care stores, and barber shops ($n$=240,312) |
| 5 | Childcare | Childcare centres ($n$=75,531) |
| 6 | Family entertainment and enrichment | Sports and recreation instruction, language schools, exam preparation and tutoring, and educational support services; music and theatre centres, zoos and botanical gardens, sports venues, bowling centres, golf courses and museums ($n$=68,696) |
| 7 | Restaurants | Restaurants and cafes ($n$=498,926) |

We leverage variation in expected income gains from the expanded CTC across counties and the timing of the benefit distributions to identify the effect of the CTC expansion on consumption in a difference-in-differences framework:

$$C_{cst}^{j} = \beta_0 + \beta_1 \text{Pov}_c + \beta_2 T_t + \beta_3 \text{PT}_t + \beta_4 T_t$$
$$\times \text{Pov}_c + \beta_5 \text{PT}_t \times \text{Pov}_c + \mathbf{X}_{cst} + \sigma_t + \sigma_s + \varepsilon_{cst}$$

where the outcome variable $C_{cst}^{j}$ is consumption in establishment type $j$, county $c$, state $s$, in month $t$. The variable Pov is a measure of the county's poverty rate and the indicator $T$ is a binary indicator of the timing of the monthly CTC payments. In our primary specification, we denote July to December 2021 (monthly payments) and March to May 2022 (lump-sum payment) as treatment months for which the variable $T$ takes the value one. We denote January and February 2022 as partially treated months (PT) given the likelihood of monthly CTC payments being used to smooth consumption in January 2022 and the payment of some initial lump-sum cheques starting at the end of February 2022. $\mathbf{X}$ is a vector of controls aimed at capturing COVID-19-related trends in local economic factors that may covary with poverty levels; we discuss these in detail in Methods. Finally, $\sigma_t$ and $\sigma_s$ represent, respectively, year–month and state fixed effects, and $\varepsilon_{cst}$ is an error term. Our primary coefficient of interest, $\beta_4$, identifies a reduced-form effect of the CTC expansion—as proxied by the county-level poverty rate—on consumption patterns. More precisely, it measures the change in visits (extensive margin) or mean spending per transaction (intensive margin) in high-poverty (or medium-poverty) versus low-poverty counties in treated months relative to untreated ones. We refer the more technical readership to Methods for more details on our data and empirical strategy.

### Consumption on extensive margin

Figure 1 presents estimates of the effect of the CTC expansion on seasonally adjusted visits to each establishment type in high- and medium-poverty counties relative to low-poverty counties. The corresponding coefficient estimates are reported in Supplementary Appendix G, Table 1. We find that the monthly and lump-sum payments both

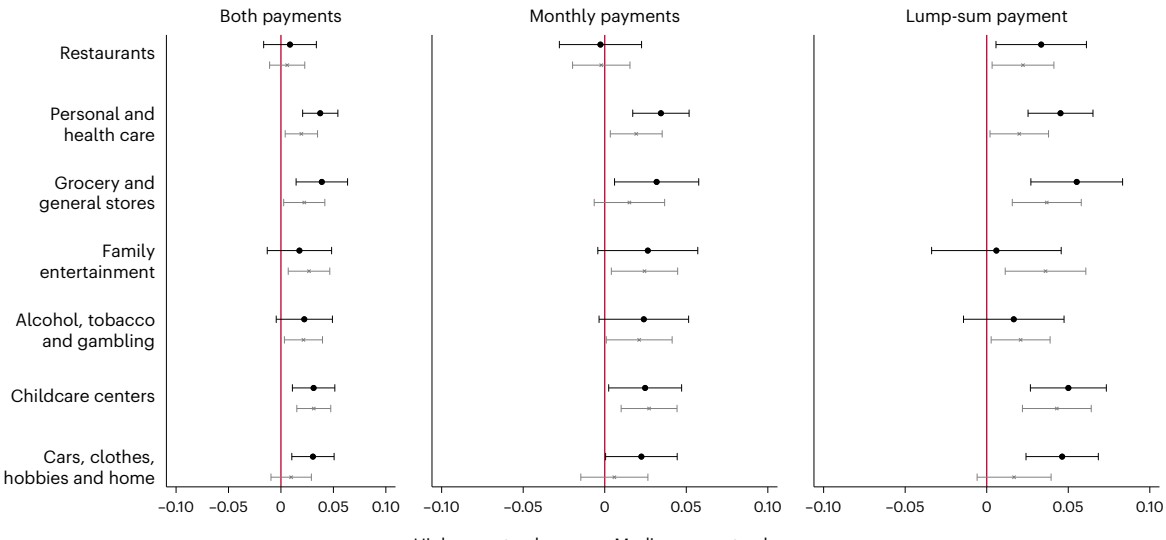

**Fig. 1 | Effects of the expanded CTC payments on seasonally adjusted visits to establishment types in high- and medium-poverty counties relative to low-poverty counties by payment type.** Estimates of coefficient $\beta_4$ from equation (1). The $x$ axis represents the estimated effect of the CTC on the percent increase in seasonally adjusted visits relative to low-poverty counties. See Table 1 for a description of establishment types. The capped horizontal bars represent 95% CIs. 'Monthly payments' refers to the effects of the monthly-distributed CTC payments between July and December 2021. 'Lump-sum payment' refers to the effects of the single, tax-time CTC payment distributed in spring 2022. Sample sizes are 51,629 (restaurants), 50,626 (personal and health care), 51,799 (general and grocery stores), 40,579 (family entertainment), 46,155 (alcohol, tobacco and gambling), 47,464 (childcare) and 50,643 (cars, clothes, hobbies and home), respectively. See Supplementary Appendix G, Table 1 for exact values of coefficients, $P$ values and CIs.

contributed to a greater increase in visits to formal childcare centres in high- and medium-poverty counties relative to low-poverty ones. In high-poverty counties, seasonally adjusted visits are estimated to have increased by 2.5% during the period of monthly payments (relative to low-poverty counties; $P = 0.031$, 95% confidence interval (CI) = 0.002 to 0.047) and by 5.0% with the larger lump-sum payment ($P < 0.0001$, 95% CI = 0.027 to 0.073). In medium-poverty counties, we estimate that visits increased by 3.1%, on average ($P < 0.0001$, 95% CI = 0.015 to 0.048), relative to low-poverty counties.

With respect to personal- and health-care establishments, we find an increase in visits for high-poverty counties, ranging from 3.4% for the monthly payments ($P < 0.0001$, 95% CI = 0.017 to 0.052) to 4.5% for the lump-sum payment ($P < 0.0001$, 95 CI = 0.025 to 0.065). For medium-poverty counties, we find an increase in visits of 1.9% ($P = 0.011$, 95% CI = 0.004 to 0.035), on average, relative to low-poverty counties.

When analysing grocery and general stores, we find an increase in visits for high-poverty counties, ranging from 3.2% for the monthly payments ($P = 0.017$, 95% CI = 0.006 to 0.057) to 5.5% for the lump-sum payment ($P < 0.0001$, 95% CI = 0.027 to 0.083). For medium-poverty counties, we find an increase in visits of 2.2% ($P = 0.023$, 95% CI = 0.003 to 0.042), on average, relative to low-poverty counties.

These three categories are those that pass the pre-trends test in both the discrete-treatment and continuous-treatment specification (see Fig. 2; Supplementary Appendix F, Fig. 1; and Supplementary Appendix D, Fig. 1, respectively). They are also robust to the inclusion of state-by-month or county fixed effects (Supplementary Appendix G).

In Supplementary Appendix D, we present results using the county-level poverty rate as a continuous treatment, rather than the poverty bins applied in Fig. 1. We use these continuous-treatment estimates to relate the magnitude of the increase in visits to the dollar value of added income due to the CTC. Back-of-the-envelope calculations suggest that, on average, a $1,000 increase in annual income due to the CTC led to a 6.7% increase in monthly visits to childcare centres. Alternatively, we find that a 10% increase in annual income due to the CTC contributed to an 11% increase in monthly visits to childcare

centres. The $1,000 increase estimate is calculated by dividing 0.177 (the association of a county's poverty rate with visits to childcare centres during the CTC treatment periods relative to before the CTC) by $2,622 (the association of a county's poverty rate with annual increase in income due to the CTC; Supplementary Appendix B), which we multiply by 1,000 to produce an estimated increase in visits due to a $1,000 increase in CTC benefits. The 10% income increase estimate is calculated by dividing the same 0.177 by 0.15 (the log gain in income associated with the CTC benefit; Supplementary Appendix B) multiplied by 10%. These elasticities allow for a simpler benchmarking of our estimates with those of other similar programmes, including the Child and Dependent Care Credit—a nonrefundable tax credit based on taxpayers' income and childcare expenses. A 10% increase in Child and Dependent Care Credit benefits has been found to increase annual paid childcare participation by 5% (ref. 17). Responses to the CTC may be slightly larger due to its refundable nature, which implies that low-income families are also largely treated.

Similar back-of-the-envelope calculations suggest that, on average, a $1,000 increase in annual income due to the CTC led to an 8.4% increase in visits to personal- and health-care establishments, and an 8.7% increase in visits to grocery and general stores. In contrast, we find no evidence that the CTC expansion led to higher-poverty counties experiencing larger seasonally adjusted visits to other establishment types, on average, over the treatment period. We note here that, although medium-poverty counties show a positive and significant effect for family entertainment and alcohol and tobacco venues, and although high-poverty counties show a positive effect for cars, clothes, hobbies and home, Fig. 2 and Supplementary Appendix F, Fig. 1 show that these effects are probably due to positive and significant pre-trends.

Figure 2 reports event-study plots of our mobility outcomes, comparing high- and mid-poverty counties with low-poverty ones. Although the standard errors are larger for any given month in this set-up compared with our baseline specification, the results are overall consistent across the two specifications. The lack of pre-policy differential trends between high- and mid-poverty and low-poverty counties

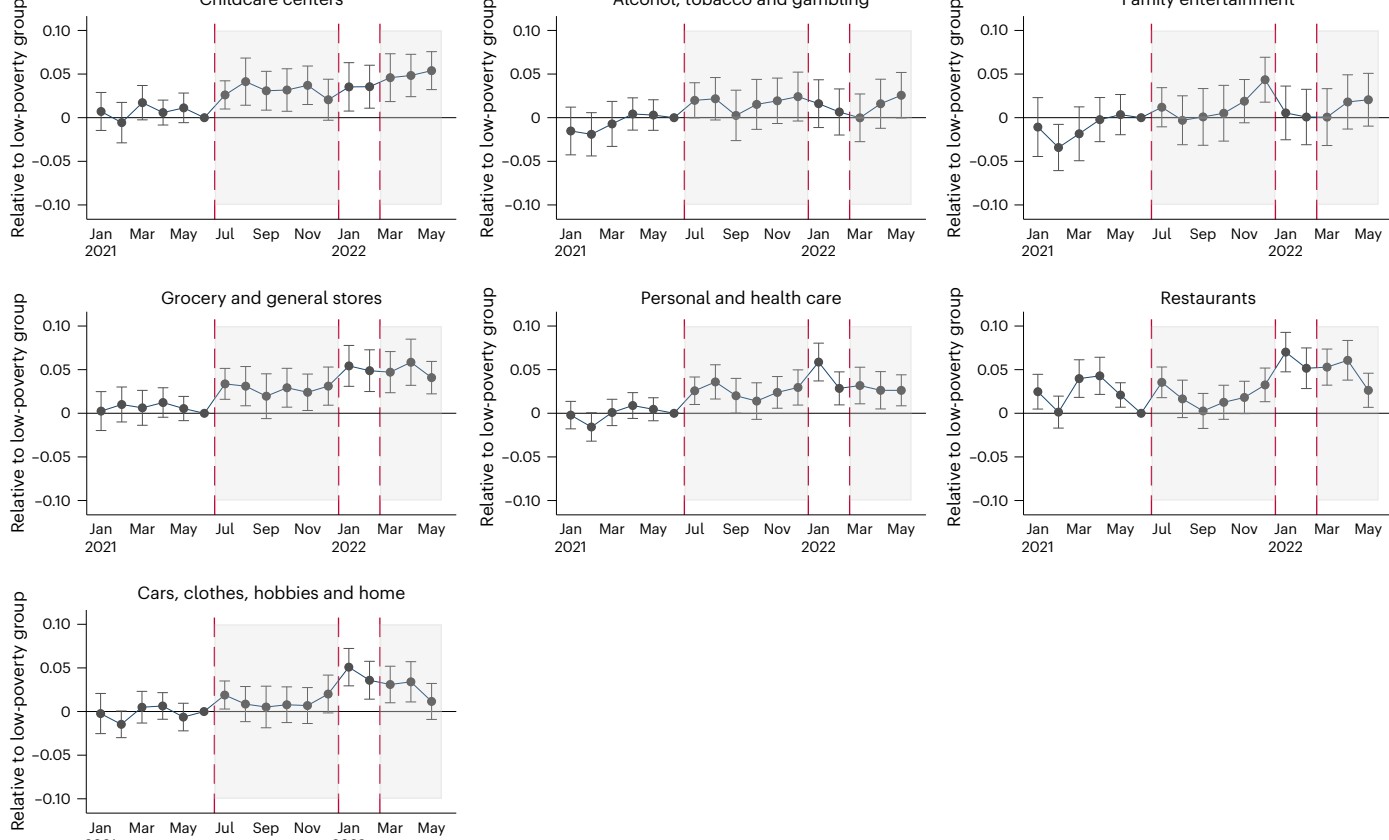

**Fig. 2 | Event-study estimates of the effect of expanded CTC on in-person visits to different establishment types in high- and medium-poverty counties relative to low-poverty counties.** The graphs show the estimated coefficients of an event-study analysis of in-person visits to different establishment types over time, comparing counties in the high- and mid-poverty terciles to counties in the lowest-poverty terciles. Estimates are relative to June 2021. The grey, shaded areas bounded by the dashed red lines represent the monthly and lump-sum CTC treatment periods, respectively. The capped horizontal bars represent 95% CIs. Sample sizes are 51,629 (restaurants), 50,626 (personal and health care), 51,799 (general and grocery stores), 40,579 (family entertainment), 46,155 (alcohol, tobacco and gambling), 47,464 (childcare) and 50,643 (cars, clothes, hobbies and home), respectively.

is also reassuring of the validity of our identifying assumption, which we will discuss in more detail below.

### Consumption on intensive margin

Figure 3 presents results of the estimated effects of the expanded CTC on spending per transaction at different establishment types. The corresponding coefficient estimates are reported in Supplementary Appendix G, Table 2.

We find evidence that the CTC payments contributed to an increase in spending per transaction at grocery and general stores but not at the other categories of interest. Specifically, the CTC payments contributed to a 1.4% higher level of mean spending at grocery and general stores ($P < 0.0001$, 95% CI = 0.006 to 0.021) in high-poverty relative to low-poverty counties, with larger effect sizes from the lump-sum payments. For medium-poverty counties, we find evidence of an increase in spending per transaction at grocery and general stores only for the lump-sum payment (1.2% increase; $P = 0.016$, 95 CI = 0.002 to 0.022). Taking the corresponding estimates from our continuous-treatment models (Supplementary Appendix D), our results imply that a $1,000 increase (alternatively, 10% increase) in annual income due to the CTC led to a 1.5% (2.4%) increase in spending per transaction at grocery and general stores. We find no evidence that the expanded CTC payments led to larger consumption volumes at other establishment types.

Figure 4 reports event-study plots of our spending outcomes, examining trends for the middle- and high-poverty counties relative to low-poverty counties. As before, estimates are consistent across specifications, although less precisely estimated in that of the event study.

We do not find consistent evidence that CTC payment frequency matters for spending decisions at the extensive margin as both the monthly and lump-sum payment types contributed to greater visits to childcare centres and personal- and health-care establishments. There is some evidence of larger increases in spending per transaction at grocery and general stores in the lump-sum payment months.

## Discussion

In providing an unconditional cash allowance to most families with children, the CTC expansion marked a historic, although temporary, shift in the social safety net's treatment of low-income families. But how did this unconditional cash allowance affect families' consumption behaviours?

Our findings suggest, first, that the CTC contributed to increases in visits to formal childcare centres, personal- and health-care establishments, and general and grocery stores. With respect to childcare, we find that a $1,000 increase in annual income due to the CTC led to a 6.7% increase in visits to childcare centres, suggesting that households could then afford formal childcare as either a complement or a substitute to informal childcare. Evidence from the Current Population Survey (CPS) indicates that employed adults in higher-poverty counties are less likely to have the option to work remotely relative to those in lower-poverty counties. As the inability to shift to remote work is likely to increase the need for childcare support to continue working, it is possible that

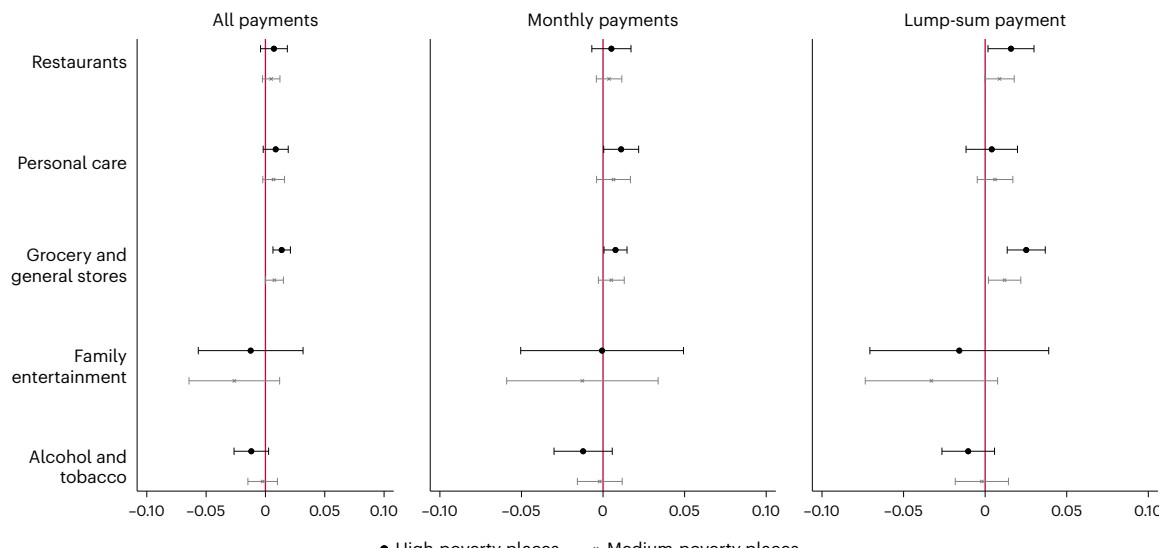

**Fig. 3 | Effects of the expanded CTC payments on spending per transaction at establishment types in high- and medium-poverty counties relative to low-poverty counties by payment type.** Estimates of coefficient $\beta_4$ from equation (1). The *x* axis represents the estimated effect of the CTC on the percent increase in mean spending per transaction relative to low-poverty counties. See Table 1 for a description of establishment types. The capped horizontal bars represent 95% CIs. Sample sizes are 24,928 (restaurants), 17,606 (personal care), 26,438 (general and grocery stores), 9,463 (family entertainment) and 13,169 (alcohol and tobacco), respectively. See Supplementary Appendix G, Table 2 for exact values of coefficients, *P* values and CIs.

**Fig. 4 | Event-study estimates of the effect of expanded CTC on spending at different establishment types in high- and medium-poverty counties relative to low-poverty counties.** The graphs show the estimated coefficients of an event-study analysis of spending at different establishment types over time, comparing counties in the high- and mid-poverty terciles to counties in the lowest-poverty tercile. Estimates are relative to June 2021. The grey, shaded areas bounded by the dashed red lines represent the monthly and lump-sum CTC treatment periods, respectively. The capped horizontal bars represent 95% CIs. Sample sizes are 24,928 (restaurants), 17,606 (personal care), 26,438 (general and grocery stores), 9,463 (family entertainment) and 13,169 (alcohol and tobacco), respectively.

this fact contributes to the larger increase in childcare consumption for high-poverty counties after the introduction of the CTC.

A number of studies have investigated employment responses to the expanded CTC, failing to detect significant responses (see ref. 18 for a review), although the long-run employment consequences of a permanently expanded CTC could be different[19]. Although an analysis of labour supply decisions is beyond the scope of this study, our findings could contribute to explaining the expanded CTC's observed lack of negative employment effects[18,20,21]. First, increases in childcare consumption are not incompatible with the lack of employment effects. Gennetian et al.[22] provide evidence that child-related cash transfers lead to increased expenditure in childcare (17%), with no associated employment effects. Second, the CTC might have lowered participation or hour constraints for some parents, while reducing employment incentives for others[1]. Data from the Annual Social and Economic Supplement of the CPS shows that, following the CTC expansion, single parents with children living in poorer counties became less likely to report being non-employed or working part-time due to childcare problems, consistent with the idea that the cash assistance provided by the CTC may have mitigated care constraints among poor families with children. Finally, the temporary nature of the expanded CTC might have reduced incentives to find a stable job in response to increased cash assistance for individuals on the margin, who could not afford formal care—or other participation-related expenses—without the more generous credit.

Second, we find that the expanded CTC contributed to higher average spending levels per transaction at grocery and general stores. The latter finding is consistent with evidence that the CTC expansion reduced food hardship among families with children[9,21].

Third, our findings complement a growing body of evidence on the behavioural effects of 'labelled cash transfers'. Governments often label cash transfers with reference to the spending target for which they are aimed. However, if there is no obligation to spend the transfers on specific items, standard economic theory predicts that labelling should have no impacts on spending. Our finding of increased visits to childcare centres and spending on child-related items echoes other evidence showing that the labelling of cash transfers matters for how they are spent[23–25]. This has implications for the ability of the government to nudge the consumption of particular goods or services.

Our findings probably understate the consumption responses to the CTC: given that approximately half of US households do not contain children, half of our underlying data sample includes childless adults. This is likely to downward bias our estimates of the CTC's effects on consumption.

Our study's use of mobile-location data and debit/credit card spending data allows us to avoid concerns on the reliability of survey-based, self-reported consumption data. Our use of more 'objective' data on consumption behaviours implies that we should be more likely than those using self-reported spending data to find higher consumption volumes at alcohol and tobacco establishments, should such consumption responses to cash transfers exist at a meaningful margin; however, we do not find consistent evidence of increased consumption at alcohol or tobacco shops, either on the intensive or extensive margins. The consistency of this finding with preliminary evidence from the Consumption Expenditure Survey[26] marks a separate contribution of our study: our more objective consumption data do not provide evidence to suggest that survey-based reports of alcohol consumption are systematically biased downwards, at least with respect to the given survey timing and context.

Moreover, although other studies have evaluated consumption responses to cash transfers using card-based data in smaller-scale, experimental settings[22,27], our analysis includes data from more than a million establishments across nearly all counties in the United States.

Our study has several limitations. First, although we can separately study consumption on the intensive and extensive margins, we lack the data to study these simultaneously for a fixed set of consumers, as one could pursue with access to individual-level credit and debit card data. Second, given that our credit and debit card spending data capture consumption on the intensive margin (mean spending value per transaction), the spending data are less useful than our mobility data for measuring consumption at institutions, such as childcare centres, where consumers do not exchange in a card-based transaction with each visit. Third, our data do not allow us to view itemized purchase receipts; thus, although we can measure increased consumption at grocery and general stores, we lack evidence on which types of item individuals are more likely to consume.

## Methods
### Data sources
We measure consumption responses to the temporary child allowance in two ways. First, we measure seasonally adjusted in-person visits to specific establishment types in each county and month between January 2019 and May 2022, providing a measure of implied consumption on the extensive margin. Second, we measure spending amounts (via debit or credit card) at most of the same establishment types between January 2021 and May 2022, providing a measure of direct consumption volumes on the intensive margin.

**In-person visits.** We measure the monthly sum of in-person visits to 1,343,200 establishments using anonymized, aggregated mobile-location data from more than 40 million mobile phones spanning 3,082 counties. The data are provided by SafeGraph and have been validated in recent research that documents school closures[28], consumption behaviours[29] and general mobility patterns[30]. SafeGraph's data are collected from third-party data sources and are not directly collected from individual persons. All third-party sources have validated that data provided to SafeGraph is acquired in a lawful manner and with terms and conditions that are explicit about anonymized data use. The data do not include any identifying information of the mobile-phone users. Instead, the in-person visits are place based: we can identify the monthly sum of visits to a given establishment. We provide further details on the SafeGraph data in Supplementary Appendix C. In Supplementary Appendix E, we show that in-person visits are highly correlated to census population counts for low-, medium- and high-poverty counties, corroborating the geographical representativeness of the mobile-location data. We have visits data for each establishment starting in January 2019; thus, although our focus is on mobility patterns throughout 2021 and 2022, we seasonally adjust our data based on visits to the same establishment–county cells in the same month in 2019. We group the 1.3 million places into 7 broad establishment types, documented in Table 1. We exclude from the analysis categories that feature a large share of places that are generally public and no cost or low cost (such as public parks), and are hence unlikely to respond to a positive income shock.

There are several advantages to the use of in-person visits as a proxy for consumption relative to the debit and credit card data. Most importantly, the place coverage of in-person visits is wider than the card-based spending data (covering 1 million more establishments), as detailed below. Moreover, the in-person visits data pick up potential customers who complete their transactions by cash, and capture visits to places such as childcare centres that are not a site of regular credit card transactions. In Supplementary Appendix E, we provide evidence that the sum of monthly in-person visits to establishments is a reliable proxy of the number of total monthly transactions at a store. However, the primary disadvantage of the in-person visits data is that these data cannot capture spending amounts associated with each transaction. It could be, for example, that the CTC income transfers do not affect visits to grocery stores but do affect the amount that a family spends on groceries once they are at the store. Thus, the in-person visits measure probably provides a conservative indication of amounts spent, and

is thus best paired with data on actual spending volumes—hence our next set of indicators.

**Spending data.** Our second source of data captures spending on the intensive margin: average debit or credit card spending per transaction at a given establishment. Data on card spending cover 316,276 establishments across 2,940 counties and are available for a subset of establishment groupings (namely, alcohol and tobacco, family entertainment, grocery and general stores, personal-care establishments and restaurants). We measure the log of mean spending per transaction among stores of an establishment type at the county–month level. We do not have access to itemized purchase receipts, hence we cannot evaluate if individuals are substituting some items for others at grocery stores. For around 60% of establishments, we also have data on mean debit/credit card spending for transactions that occur online; given the imperfect coverage and generally low values of online spending, we do not include these in our primary analysis. In additional analyses, we find positive but highly imprecisely estimated differences in online spending amounts for high- and medium-poverty counties relative to low-poverty counties after the CTC expansion relative to before at any of our establishment types. The county-level means are weighted by the total number of transactions at each individual store. Unlike for our mobility data, we do not have card-spending data for establishments in 2019; thus we cannot apply the same seasonal adjustment. We do not use any spending data from 2020 in our analysis and do not seasonally adjust to a 2020 baseline, given that spending patterns for 2020 are unlikely to be a reliable baseline. We instead rely on an assumption that seasonal variation in spending patterns in 2021 will affect high-, medium- and low-poverty counties alike.

**Unit of analysis.** Although we can identify the census tract and block group of each establishment type, we aggregate all visits to the county level given that (1) consumption decisions do not primarily occur within tracts; (2) most tracts are small enough that they have only one (if any) of a given establishment type; (3) aggregating to county helps us account for substitution among places in nearby but separate census tracts; (4) we cannot reliably estimate income gains from the expanded CTC below the county level; and (5) public data on COVID-19 cases and unemployment rates, potential confounding factors, are only available at the county level (or higher). We aggregate consumption patterns to the year, month, county and establishment-type levels; thus, for each establishment type, we have a sample size of 52,394 county–months (3,082 counties by 17 months).

**Seasonal adjustments.** Consumption patterns may increase in certain months, such as the holiday period of December, independent of the CTC. We seasonally adjust our in-person visits data for each month, county and establishment type in 2021 and 2022 to account for seasonality. Specifically, we measure the percent change in consumption for the 2021 or 2022 month relative to the 2019 month (our pre-pandemic reference point) for the same county and establishment type. Thus, a value equal to 0 represents no change in consumption from our reference period, whereas a value of 0.50 represents a 50% increase in visits relative to the reference period. We bottom- and top-code changes at −1 and 1, respectively. As mentioned before, we cannot seasonally adjust our spending data.

**Comparison with alternative data sources.** The CES is another data source that could be used to examine changes in consumption and spending in response to the CTC[26]. However, the CES asks households about their aggregate expenditures in the 3 months before their interview, thus there are CES interview months where the reporting period included months where families were and were not receiving the monthly CTC payments (for example, when reporting on spending between May and July 2021), and the same is true for the lump-sum

payments. Although adjustments for partially treated quarters can be applied, the quarterly reporting structure in the CES data nonetheless complicates attempts to estimate the CTC's effects on consumption behaviours, particularly across the two payment (monthly and lump-sum) types. The CES data are also limited in that they are survey based and prone to bias or recall error, as discussed previously. Our data, in contrast, are not vulnerable to these concerns, although they are limited in that we do not observe individual-level data. Comparing our findings from objective, location-based consumption data (particularly for sensitive consumption categories, such as alcohol and tobacco) with findings from studies that use self-reported consumption behaviours is thus an added contribution of our study: overlapping findings would suggest that recall error and reporting bias may be less of a concern in interpreting survey-based consumption data than prior research suggests[31], an important insight.

### Identification strategy

We leverage variation in expected income gains from the expanded CTC across counties and the timing of the benefit distributions to identify the effect of the CTC expansion on consumption in a difference-in-differences framework. As validated in Supplementary Appendix B and previously discussed, we use a county's poverty rate as our proxy for CTC gains to maximize county coverage. Equation (1) details our empirical specification:

$$C_{cst}^{j} = \beta_0 + \beta_1 \text{Pov}_c + \beta_2 T_t + \beta_3 \text{PT}_t + \beta_4 T_t$$
$$\times \text{Pov}_c + \beta_5 \text{PT}_t \times \text{Pov}_c + \mathbf{X}_{cst} + \sigma_t + \sigma_s + \varepsilon_{cst} \tag{1}$$

Our outcome variable $C_{cst}^{j}$ is consumption in establishment type $j$, county $c$, state $s$, in month $t$. $C_{cst}^{j}$ will denote either (1) our seasonally adjusted measure of in-person visits (extensive margin), where $C_{cst}^{j}$ is the percent change in number of visits between month $t$ of year 2021 or 2022 and the same month in 2019; or (2) the log of mean spending per transaction (intensive margin). Our treatment variable Pov is a measure of the county's poverty rate, measured using 5 year averages of county poverty rates from 2014 to 2018, as estimated in the ACS. We avoid any data from 2019–2021 given biased response rates and potentially altered poverty rates due to data collection challenges during the COVID-19 pandemic. In our empirical analysis, we use two different versions of our treatment variable. In our main specification, Pov indicates a set of indicators for the tercile of the county's poverty rate in the national distribution (low, medium or high poverty). The variable Pov thus corresponds to separate indicator variables for medium and high poverty, with low poverty being the excluded category. Supplementary Appendix H, Fig. 1 presents a map of all counties in the sample by their poverty status. In Supplementary Appendix D, we also present results from a version of equation (1) in which the variable Pov is a continuous measure of the poverty rate.

Our treatment indicator, $T$, is a binary indicator of the timing of the monthly CTC payments. In our primary specification, we denote July to December 2021 (monthly payments) and March to May 2022 (lump-sum payment) as treatment months for which the variable $T$ takes the value of one. We denote January and February 2022 as partially treated months (PT) given the likelihood of monthly CTC payments being used to smooth consumption in January 2022 and the payment of some initial lump-sum cheques starting at the end of February 2022. In Supplementary Appendix G, we assess the robustness of our estimates to counting January and February 2022 as treated or untreated months.

We acknowledge that our estimates of the timing of lump-sum payments is less precise than the standardized timing of the monthly payments, as the IRS distributes lump-sum payments beginning in late February as individuals file their taxes. In recent years, the IRS has distributed around half of EITC benefits (a refundable tax credit for which we have available data on payment timing) in the final week of February, whereas approximately three-fourths are paid out by the

end of March and approximately 90% are paid out by the end of April[32]. We show in Supplementary Appendix G that our results are robust to including February into the treatment group for the lump-sum payments. In addition, we present event-study plots of the effect of the expanded CTC on consumption for each month, which provide a transparent illustration of the timing of responses.

The interaction of the 'full' treatment timing and treatment intensity, captured by $\beta_4$, is our primary coefficient of interest. Multiplied by 100, it captures the percent change in visits (extensive margin) or mean spending per transaction (intensive margin) in high-poverty (or medium-poverty) versus low-poverty counties in treated months relative to untreated ones. Note that as we separately control for the interaction between the poverty rate and partially treated months ($\beta_5$), our coefficient $\beta_4$ captures consumption patterns in our primary treatment months relative to pre-treatment months (January to June 2021). As such, the coefficient identifies a reduced-form effect of the CTC expansion—as proxied by the county-level poverty rate—on consumption patterns. When presenting our results, we report reduced-form effects for all treatment months (which we label 'all payments' as they include both the monthly and lump-sum payments), and then separately for July to December 2021 (monthly payments) and for March to May 2022 (lump-sum payments).

$X$ is a vector of controls including the log of population density in the county, the number of monthly COVID-19 cases in the county, state–month indicators for the availability of Supplemental Nutrition Assistance Program Emergency Allotments (which allow states to increase Supplemental Nutrition Assistance Program benefit levels to the maximum benefit amount), state–month indicators for the availability of expanded unemployment benefits (which some states began withdrawing in the summer of 2021), the county–month unemployment rate from the Bureau of Labor Statistics, the county–quarter log of average weekly wages among all workers from the Quarterly Census of Employment and Wages, and an indicator of state–month policies on COVID-19 mobility restrictions from the Oxford COVID-19 Government Response Tracker. Specifically, this final indicator includes the Government Response Tracker's Containment and Health Index measure, an index that combines 13 indicators of state-level polices related to workplace closures, cancellation of public events, restrictions on public gatherings, closures of public transport, stay-at-home requirements, public information campaigns, restrictions on internal movements, international travel controls, testing policy, extent of contact tracing, face coverings and vaccine policy. The index takes the mean score across the normalized distributions for each metric to produce a combined index valued between 0 and 100, with a higher score implying stricter containment measures. These variables are meant to control for COVID-19-related factors that may be systematically associated with poverty bins in the post-treatment period. We also include seasonally adjusted visits to public schools among our controls. The latter, which we will show are unaffected by CTC payments, is meant to control for any place- or time-based error in the mobile-location or card spending data. We are unable to control for county-level variation in gains from the childless EITC expansion provided upon tax filing in 2022, but we document in Supplementary Appendix B that small differences in associated gains across poverty terciles are unlikely to meaningfully affect our findings.

Finally, $\sigma_s$ and $\sigma_t$ represent, respectively, state and year–month fixed effects, and $\varepsilon_{cst}$ is an error term. We estimate equation (1) separately for each of the seven establishment types $j$ listed in Table 1, but for ease of notation we do not index all parameters to $j$ in equation (1). We cluster standard errors at the state level. We weight our primary estimates by the square root of the county's population size. We present unweighted results in Supplementary Appendix G. We also replicate our core findings, including state-by-month and county fixed effects, in Supplementary Appendix G. In case our set of controls did not adequately account for COVID-19-related factors that may affect mobility

and consumption, those factors should be implicitly controlled for in these alternative specifications.

Our key identification assumption is that, absent the CTC expansion, trends in extensive- and intensive-margin consumption in each category would have evolved similarly across poorer and richer counties over time. We also require that there are no spillovers in consumption across counties with different poverty rates: if CTC recipients respond to the tax credit by consuming more in lower- or higher-poverty counties than in their own county, then our estimate of $\beta_4$ would be biased. We provide supporting evidence for both of these assumptions below.

### Measurement, identification and robustness checks
In the next paragraphs, we discuss a set of tests performed to assess the reliability of our measures of consumption, the validity of our identification assumptions and the robustness of our findings to alternative specification choices.

**Measurement validation.** We first evaluate the reliability of in-person visits as a proxy of consumption. We do so by comparing our measure of in-person visits to the total number of transactions made with debit or credit cards in the same establishment type, county and month in 2021. For this analysis, we can only compare the establishment types for which debit or credit card data are regularly available (Table 1). As Supplementary Appendix E, Fig. 1 shows, we see very strong, positive associations of within-county variation in visits and customer transactions across all establishment types. We conclude that our data on in-person visits—which covers 1 million more establishments than the card-based spending data—are a generally reliable proxy of consumption. Recall that we still use card-based data on spending volumes (covering 316,276 establishments) for our separate analyses regarding consumption on the intensive margin.

Second, we provide evidence of the representativeness of our mobility data. In Supplementary Appendix E, Fig. 2, we start by showing that our in-person visit sums are very strongly, positively associated with census population counts for each of our three poverty groups. However, being based on mobile location, our in-person visits data are only representative of consumption among the subpopulation of smartphone owners. If smartphone ownership was systematically correlated with the poverty rate, or was differentially correlated with household income across lower- versus higher-poverty counties, then our results could be biased by differences in the composition of the population covered by mobile-location data across treated and control units. Using data from the ACS, we document that there is little variation in the rates of smartphone ownership across higher-poverty and lower-poverty counties (Supplementary Appendix E, Fig. 3). However, the ACS data also suggest that low-household-income adults in high-poverty counties are up to 7% less likely to own a smartphone relative to low-household-income adults in low-poverty counties (Supplementary Appendix E, Fig. 4). If families with and without smartphones have similar preferences for consumption (of childcare and other items), this implies that our results probably understate the CTC's impact on consumption as we would be missing relatively more information for the group that is most likely to benefit from the policy change. If, instead, families without smartphones have different consumption preferences, for instance, are averse to the use of childcare, then our estimates would overstate the effect of interest.

Third, to assess the reliability of our mobile-location data, we conduct two placebo tests that evaluate whether the CTC expansion is associated with changes in seasonally adjusted visits to (1) public K–12 schools and (2) religious institutions, such as churches, mosques and synagogues. The CTC should not meaningfully affect visits to public schools as school enrolment is mandatory for students up to age 18 and monthly variation in attendance among schools in a given county is unlikely to be driven by families' income fluctuations. Similarly, we

do not anticipate that CTC gains would meaningfully affect individuals' decisions to attend a place of worship. Thus, finding an association between our treatment and changes in visits to schools or religious institutions would suggest notable selection or measurement error in our consumption data. As shown in Supplementary Appendix E, Figs. 5 and 6, our placebo tests fail to detect any systematic association between school visits (or place-of-worship visits) and the poverty indicators.

**Identification tests.** As a complement of our event-study analysis, we further probe the validity of our parallel-trends assumption by running a pre-trends analysis of in-person visits and spending per transaction across all seven establishment types. Specifically, we estimate a version of equation (1) that restricts the time span to pre-treatment months (January to June 2021) and substitutes the treatment timing indicator with a month time trend. Supplementary Appendix F, Fig. 1 shows that nearly all establishment types pass our pre-trends assumption for high-poverty counties. The only exception is a positive, significant pre-trend for spending on personal-care items in high-poverty counties relative to low-poverty ones.

In Supplementary Appendix F, Fig. 2, we show a version of the event-study plots for our in-person visits estimates, extending the pre-treatment period to July 2020. Our primary findings on visits to childcare centres still hold. However, as the second half of 2020 was more fraught with COVID-19 cases and associated policy interventions, we prefer to use January to June 2021 as a pre-treatment period.

For identification, we also require that there be no meaningful spillovers in consumption across counties with different poverty rates. To test the validity of this assumption, we present an alternative set of results that restricts our sample to counties that are in the same poverty tercile of the majority of their neighbouring (contiguous) counties. Restricting the analysis to this sample, which covers 64% of all counties, mitigates the concern that our estimates of increased consumption in poorer counties may be driven by increased spending in those counties by individuals residing in neighbouring richer counties. The findings reported in Supplementary Appendix F, Figs. 3 and 4 are very similar to our main estimates.

Finally, we present evidence of coverage of the CTC benefits across the household income distribution by county poverty rate. Using the Annual Social and Economic Supplement of the CPS for 2022 (covering calendar year 2021), Supplementary Appendix B, Fig. 2 reports the estimated difference in take-up of the monthly CTC payments in medium- and high-poverty counties, relative to low-poverty ones, by within-county household income decile. We do not detect significant differences in CTC take-up across the entire within-county household income distribution.

**Robustness checks.** We perform several additional analyses to assess the robustness of our findings, starting from the definition of treated months. In Supplementary Appendix G, Figs. 1 and 2, we report estimates of a version of equation (1) in which we bundle together monthly and lump-sum payments into a single treatment dummy, and assess the robustness of our estimates to counting the 'partially treated' months (January to February 2022) as treated (left) or not treated (right). The results are robust to these alternative definitions of the partially treated months. If anything, our estimates of in-person visits are amplified when including January to February 2022 among treated months, consistent with forward-looking behaviour and consumption smoothing. In Supplementary Appendix G, Figs. 3 and 4, we report results for a version of equation (1) in which we define months from February to May 2022 (instead of March to May 2022) as the lump-sum payment. The results are very similar to our baseline estimates.

We then turn to assessing the robustness of our primary findings to a set of alternative specifications. Supplementary Appendix G, Table 3 reports results for in-person visits to childcare centres based on the

following models, from top to bottom: (1) our baseline estimates from equation (1), which include state and year–month fixed effects, and a broad set of controls; (2) our baseline estimates but also with interacted state-by-year or month fixed effects; and (3) our baseline estimates but also with county fixed effects. Our findings regarding increased extensive-margin consumption at childcare centres hold across all models. Supplementary Appendix G, Fig. 10 shows the corresponding event-study analyses.

Supplementary Appendix G, Tables 5–7 (and Supplementary Figs. 11–13) report the same battery of specifications for consumption at personal- and health-care establishments and grocery and general stores. Our findings of increased in-person visits and increased spending per transaction at grocery and general stores (under lump-sum payments) are robust to different specification choices.

It is important to note that, as our definition of the poverty-rate terciles is based on the national distribution of the poverty rate, the inclusion of state fixed effects is likely to soak up most of the variation we rely on for identification. To allow for the inclusion of state fixed effects while retaining variation in poverty rates, we estimate a version of equation (1) that includes state fixed effects and uses poverty terciles based on the state-level distribution of county poverty rates. This specification allows us to control more finely for any state-level policies or shocks that may be correlated with the poverty index and—at the same time—may have affected individual mobility or spending patterns over the period analysed. In our national analysis, some states (such as Alabama) primarily feature high-poverty counties, as even their wealthier counties tend to have higher poverty rates relative to the national distribution; in other states (such as Vermont), most counties are classified as low poverty (Supplementary Appendix H, Fig. 1). In this alternative within-state specification, we measure each county's poverty status relative to other counties within the same state. For example, Alabama would have the same share of 'low-poverty' counties as Vermont (roughly one-third of either state's counties; Supplementary Appendix H, Fig. 2). As shown in the bottom of Table 3 in Supplementary Appendix G, higher-poverty counties still show greater seasonally adjusted visits to childcare centres relative to lower-poverty counties, with effect sizes of magnitude comparable with our main estimates. Supplementary Appendix G, Tables 4–6 report similar estimates for other relevant establishment types.

Supplementary Appendix G, Figs. 5 and 6 present unweighted estimation results, where each county counts equivalently regardless of population size. Results are not meaningfully different.

Our primary results are based on all counties for which we have some data, even if some counties do not feature an establishment of a given type. In Supplementary Appendix G, Figs. 7 and 8, we restrict the analysis to the subset of counties that have mobility or spending data for at least one of each establishment type within their borders to achieve consistent county coverage across estimates spanning all establishment types. In practice, this restriction removes 950 counties (covering 3.7% of the US population) with smaller population sizes and densities, but higher poverty rates (Supplementary Appendix G, Table 7). The results from these restricted analyses are consistent with our primary findings, although slightly attenuated and less precisely estimated, as one would expect given the sample restriction.

Finally, in Supplementary Appendix G, Fig. 9, we further disaggregate one establishment type—clothing—in its subcategories to consider another dimension of spending on children that we can observe in our data. We show that higher-poverty counties were more likely to increase (seasonally adjusted) visits to 'children's and infant's clothing' and 'family clothing' stores relative to all other clothing stores (clothing stores labelled as men's, women's, accessories or other) as a result of the lump-sum CTC payment. Thus, even when analysing within-category variation in types of spending for clothing, we find evidence from the lump-sum payment of prioritization of family-oriented spending as a result of the child allowance.

## Reporting summary

Further information on research design is available in the Nature Portfolio Reporting Summary linked to this article.

## Data availability

The data that support the findings of this study are available from SafeGraph, a private company that regulates registered users' distribution of its data. For this reason, we cannot publish the underlying SafeGraph dataset online. The mobility and debit/credit card spending data are both available from SafeGraph after registering for data access at https://www.safegraph.com/. The replication code in our online files can convert the raw data into a usable dataset.

## Code availability

The authors have made replication code available through the Open Science Framework (OSF) repository. This can be accessed at https://osf.io/gd5wu/. The code was generated using Stata v.17.

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

## Acknowledgements

Z.P. acknowledges funding from the Bill & Melinda Gates Foundation and the European Union (ERC Starting Grant, ExpPov, number 101039655; Z.P.). Views and opinions expressed are, however, those of the authors only and do not necessarily reflect those of the European Union or the European Research Council; neither the European Union nor the granting authority can be held responsible for them. The funders had no role in study design, data collection and analysis, decision to publish or preparation of the article.

## Author contributions

Z.P. led analysis and writing. G.G. contributed to methods, analysis and writing. E.K.L. led data collection. S.C. contributed to analysis.

## Competing interests

The authors report no competing interests.

## Additional information

**Correspondence and requests for materials** should be addressed to Zachary Parolin.

# Reporting Summary

## Statistics

For all statistical analyses, confirm that the following items are present in the figure legend, table legend, main text, or Methods section.

| n/a | Confirmed | |
|---|---|---|
| ☐ | ☒ | The exact sample size (*n*) for each experimental group/condition, given as a discrete number and unit of measurement |
| ☐ | ☒ | A statement on whether measurements were taken from distinct samples or whether the same sample was measured repeatedly |
| ☐ | ☒ | The statistical test(s) used AND whether they are one- or two-sided *Only common tests should be described solely by name; describe more complex techniques in the Methods section.* |
| ☐ | ☒ | A description of all covariates tested |
| ☐ | ☒ | A description of any assumptions or corrections, such as tests of normality and adjustment for multiple comparisons |
| ☐ | ☒ | A full description of the statistical parameters including central tendency (e.g. means) or other basic estimates (e.g. regression coefficient) AND variation (e.g. standard deviation) or associated estimates of uncertainty (e.g. confidence intervals) |
| ☐ | ☒ | For null hypothesis testing, the test statistic (e.g. *F*, *t*, *r*) with confidence intervals, effect sizes, degrees of freedom and *P* value noted *Give P values as exact values whenever suitable.* |
| ☒ | ☐ | For Bayesian analysis, information on the choice of priors and Markov chain Monte Carlo settings |
| ☒ | ☐ | For hierarchical and complex designs, identification of the appropriate level for tests and full reporting of outcomes |
| ☒ | ☐ | Estimates of effect sizes (e.g. Cohen's *d*, Pearson's *r*), indicating how they were calculated |

*Our web collection on statistics for biologists contains articles on many of the points above.*

## Software and code

Policy information about availability of computer code

| | |
|---|---|
| Data collection | No software was used for unique data collection. We imported the data from SafeGraph into Stata version 17. |
| Data analysis | We conduct all analyses using Stata version 17. |

For manuscripts utilizing custom algorithms or software that are central to the research but not yet described in published literature, software must be made available to editors and reviewers. We strongly encourage code deposition in a community repository (e.g. GitHub). See the Nature Portfolio guidelines for submitting code & software for further information.

## Data

Policy information about availability of data

All manuscripts must include a data availability statement. This statement should provide the following information, where applicable:
- Accession codes, unique identifiers, or web links for publicly available datasets
- A description of any restrictions on data availability
- For clinical datasets or third party data, please ensure that the statement adheres to our policy

We provide the code to replicate our findings in an online repository stored at  https://osf.io/gd5wu/. The data that support the findings of this study are available from SafeGraph, a private company that regulates registered users' distribution of its data. For this reason, we cannot publish the underlying SafeGraph dataset online. The mobility and debit/credit card spending data are both available from SafeGraph after registering for data access at https://www.safegraph.com/. The replication code in our online files can convert the raw data into a usable dataset.

# Research involving human participants, their data, or biological material

Policy information about studies with <u>human participants or human data</u>. See also policy information about <u>sex, gender (identity/presentation), and sexual orientation</u> and <u>race, ethnicity and racism</u>.

| | |
|---|---|
| Reporting on sex and gender | Sex- or gender-based analyses are not possible within our analysis, as the mobility and card spending data are aggregated to the place level. We cannot disaggregate consumption patterns by sex or gender. |
| Reporting on race, ethnicity, or other socially relevant groupings | Race/ethnicity-based analyses are not possible within our analysis, as the mobility and card spending data are aggregated to the place level. |
| Population characteristics | Our place-based data covers nearly all counties in the United States during 2019 and 2021. |
| Recruitment | We rely on secondary data and did not recruit participants for the study. |
| Ethics oversight | The study relies on the use of secondary data that is legally collected, aggregated and anonymized upon collection, and made freely available to academic researchers. Ethics approval was not required. |

Note that full information on the approval of the study protocol must also be provided in the manuscript.

# Field-specific reporting

Please select the one below that is the best fit for your research. If you are not sure, read the appropriate sections before making your selection.

☐ Life sciences  ☒ Behavioural & social sciences  ☐ Ecological, evolutionary & environmental sciences

For a reference copy of the document with all sections, see nature.com/documents/nr-reporting-summary-flat.pdf

# Behavioural & social sciences study design

All studies must disclose on these points even when the disclosure is negative.

| | |
|---|---|
| Study description | This study uses quantitative data from SafeGraph to estimate mobility and spending responses to the expansion of the Child Tax Credit in 2021. |
| Research sample | The SafeGraph sample includes more than 40 million mobile device users across the United States. The SafeGraph data collected is anonymous at point of collection. As detailed in the manuscript, the mobile devices are dispersed randomly across the U.S. and align with Census population counts by county and state. We use the anonymized, aggregated data that SafeGraph provides free-of-charge to researchers. The sample is meant to represent the population of U.S. residents, relevant because this is the group affected by the treatment that we study (the Child Tax Credit expansion). |
| Sampling strategy | The sample includes mobile devices across the U.S. that agree to (or do not elect not to) share their anonymized location information with third-parties from whom SafeGraph collects data. The SafeGraph data collected is anonymous at point of collection. Data made available for researchers are anonymous and aggregated to the month and point-of-interest (e.g. shops, schools, child care centers, etc.) levels. Samples are non-random and sample sizes are determined based on the full availability of data providers to SafeGraph's aggregation of mobility and spending data. |
| Data collection | The sample includes mobile devices across the U.S. that agree to (or do not elect not to) share their anonymized location information with third-parties from whom SafeGraph collects data. The SafeGraph data collected is anonymous at point of collection. Data made available for researchers are anonymous and aggregated to the month and point-of-interest (e.g. shops, schools, child care centers, etc.) levels. SafeGraph collects the data from third-party data providers. |
| Timing | Data collection is continuous among SafeGraph devices. The data made public are aggregates over the course of all days in a given month. The data run from January 1, 2019, to June 1, 2022. |
| Data exclusions | Individuals without mobile devices or who elect not to share their data with third-party sources are excluded from the SafeGraph data universe. As discussed in the manuscript, non-participants do not appear to bias the data; SafeGraph mobile device counts are strongly and positively correlated with Census population counts across county and state. |
| Non-participation | Individuals without mobile devices or who elect not to share their data with third-party sources are excluded from the SafeGraph data universe. As discussed in the manuscript, non-participants do not appear to bias the data; SafeGraph mobile device counts are strongly and positively correlated with Census population counts across county and state. |
| Randomization | Randomization was not necessary for this study. We control through covariates at the county level using data from the U.S. Census Bureau and Opportunity Insights, as detailed in the manuscript. |

# Reporting for specific materials, systems and methods

We require information from authors about some types of materials, experimental systems and methods used in many studies. Here, indicate whether each material, system or method listed is relevant to your study. If you are not sure if a list item applies to your research, read the appropriate section before selecting a response.

## Materials & experimental systems

| n/a | Involved in the study |
|-----|------------------------|
| ☒ | Antibodies |
| ☒ | Eukaryotic cell lines |
| ☒ | Palaeontology and archaeology |
| ☒ | Animals and other organisms |
| ☒ | Clinical data |
| ☒ | Dual use research of concern |
| ☒ | Plants |

## Methods

| n/a | Involved in the study |
|-----|------------------------|
| ☒ | ChIP-seq |
| ☒ | Flow cytometry |
| ☒ | MRI-based neuroimaging |

## Plants

**Seed stocks**
*Report on the source of all seed stocks or other plant material used. If applicable, state the seed stock centre and catalogue number. If plant specimens were collected from the field, describe the collection location, date and sampling procedures.*

**Novel plant genotypes**
*Describe the methods by which all novel plant genotypes were produced. This includes those generated by transgenic approaches, gene editing, chemical/radiation-based mutagenesis and hybridization. For transgenic lines, describe the transformation method, the number of independent lines analyzed and the generation upon which experiments were performed. For gene-edited lines, describe the editor used, the endogenous sequence targeted for editing, the targeting guide RNA sequence (if applicable) and how the editor was applied.*

**Authentication**
*Describe any authentication procedures for each seed stock used or novel genotype generated. Describe any experiments used to assess the effect of a mutation and, where applicable, how potential secondary effects (e.g. second site T-DNA insertions, mosiacism, off-target gene editing) were examined.*

