## [Peer Review File · Nature Human Behaviour]

Peer Review Information

Journal: Nature Human Behaviour

Manuscript Title: Consumption Responses to an Unconditional Child Allowance in the United States

Corresponding author name(s): Zachary Parolin

Reviewer Comments & Decisions:

Decision Letter, initial version:

10th March 2023

Dear Dr. Parolin,

Thank you once again for your manuscript, entitled "Consumption Responses to an Unconditional Child Allowance in the United States", and for your patience during the peer review process.

Your Article has now been evaluated by 3 referees. You will see from their comments copied below that, although they find your work of potential interest, they have raised quite substantial concerns. In light of these comments, we cannot accept the manuscript for publication, but would be interested in considering a revised version if you are willing and able to fully address reviewer and editorial concerns.

We hope you will find the referees' comments useful as you decide how to proceed. If you wish to submit a substantially revised manuscript, please bear in mind that we will be reluctant to approach the referees again in the absence of major revisions. We are committed to providing a fair and constructive peer-review process. Do not hesitate to contact us if there are specific requests from the reviewers that you believe are technically impossible or unlikely to yield a meaningful outcome.

To guide the scope of the revisions, the editors discuss the referee reports in detail within the team, including with the chief editor, with a view to (1) identifying key priorities that should be addressed in revision and (2) overruling referee requests that are deemed beyond the scope of the current study. We hope that you will find the prioritised set of referee points to be useful when revising your study. Please do not hesitate to get in touch if you would like to discuss these issues further.

In particular, your revision must address the following (as well as all other reviewer comments):

- 1) Address reviewer concerns about your regression framework, including fixed effects and clustered standard errors as recommended by the reviewers.

- 2) Address reviewer concerns regarding the comprehensiveness of the covariate set included and the potential for a number of other variables to confound the estimates.
- 3) Ensure that your methodological choices are clear and fully justified, and that the pattern of results is clearly explained.
- 4) Include a thorough discussion of the limitations of the data and analyses, and transparently discuss the implications of these.

Please note in particular that for manuscripts where key results are null, we require the following:

- Evidence that the study is sufficiently powered to detect the smallest theoretically or pragmatically meaningful effect
- Bayes Factors or equivalence tests to interpret the null results
- Appropriate language to describe the results. (There is no statistical test that can demonstrate absence of an effect. Statements such as 'There is no difference between x and y.' or 'X does not affect Y.' must be revised to read 'We found [no/little] credible evidence of a difference between x and y.' or 'We found [no/little] credible evidence that X affects Y.')

If you wish to submit a suitably revised manuscript, we would hope to receive it within 6 months. I would be grateful if you could contact us as soon as possible if you foresee difficulties with meeting this target resubmission date.

- Include a "Response to the editors and reviewers" document detailing, point-by-point, how you addressed each editor and referee comment. If no action was taken to address a point, you must provide a compelling argument. When formatting this document, please respond to each reviewer comment individually, including the full text of the reviewer comment verbatim followed by your response to the individual point. This response will be used by the editors to evaluate your revision and sent back to the reviewers along with the revised manuscript.
- Highlight all changes made to your manuscript or provide us with a version that tracks changes.

[REDACTED]

Thank you for the opportunity to review your work. Please do not hesitate to contact me if you have any questions or would like to discuss the required revisions further.

Sincerely,
Aisha

Aisha Bradshaw, PhD
Senior Editor
Nature Human Behaviour

Reviewer expertise:

Reviewer #1: economics, tax policy

Reviewer #2: public policy, tax credits

Reviewer #3: family economics

REVIEWER COMMENTS:

Reviewer #1:

Remarks to the Author:

Interesting and important question with great data. I appreciate going beyond the standard CES or Pulse data.

Are you accounting for EITC refunds also happening in March-May?

Many purchases were made online and not in person in 2021. If anything, I'd expect more purchases online for higher income people, which could drive up the relative number of in-person purchases that you'll attribute to the CTC. Is this a threat to identification?

How do your results compare with "Spending Responses to High-Frequency Shifts in Payment Timing: Evidence from the Earned Income Tax Credit" by Aladangady et al? Be sure to cite this paper along with Goodman-Bacon & McGranahan, 2008.

Paragraph on page 15 is too long and dense.

Reviewer #2:

Remarks to the Author:

The study "Consumption Responses to an Unconditional Child Allowance in the United States" examines consumption responses to the 2021 Child Tax Credit expansion using anonymized mobile-location data and debit/credit card data. This is an innovative study design and the authors

well motivate the study's aims. I have several suggestions that I hope might improve the manuscript.

1) The manuscript would be improved if the authors more thoroughly introduced and justified some of their methodological decisions. In particular, the authors should explain why they chose March-May 2022 as "treatment months" for analyzing lump-sum CTC effects on consumption. While issuance timing of the monthly 2021 CTC payments around the 15th of each month from July to December 2021 is clear, there is less certainty about the issuance and receipt timing of lump-sum CTC.

2) Similarly, the authors should also discuss measurement error in their proxy for lump-sum CTC "treatment" (represented by an indicator variable for "lump-sum treatment" months) given that there is considerable variability in the timing and magnitude of lump-sum tax credit receipt, which goes uncaptured by an indicator variable. Several studies present data showing that the timing of lump-sum tax credit receipt is far from uniform around tax time (Jones & Michelmore; Aladgady et al., 2018) and is correlated with household income (Maag, Roll, Oliphant, 2016). The authors should consider how this source of measurement error in their treatment variable could influence their results. (This issue is less concerning for treatment T, a binary indicator of the timing of monthly CTC receipt, since the monthly CTC payments were received around the 15th of each month, and thus, were fairly uniform in terms of both timing and total monthly outlays over the 6-month period).

Jones, L. E., & Michelmore, K. (2019). Timing is money: Does lump-sum payment of the earned income tax credit affect savings and debt?. *Economic Inquiry*, 57(3), 1659-1674.

Aladgady, A., Aron-Dine, S., Cashin, D., Dunn, W., Feiveson, L., Lengermann, P., ... & Sahm, C. (2018, January). High-frequency spending responses to the Earned Income Tax Credit. In *Proceedings. Annual Conference on Taxation and Minutes of the Annual Meeting of the National Tax Association*(Vol. 111, pp. 1-10). National Tax Association.

Maag, E., Roll, S., & Oliphant, J. (2022). Delaying tax refunds for earned income tax credit and additional child tax credit claimants. Tax Policy Center.

3) The authors do not include county fixed-effects in their regression models, which seem appropriate in order to control for any time-invariant, unmeasured characteristics of the county. (I am especially concerned about unobserved factors that are correlated with poverty status, which the authors use as a proxy for CTC gains.) Can the authors describe why county fixed-effects were omitted from their models? Ideally, the authors could show that their results are robust to the inclusion of county fixed-effects.

4) For identification, the authors "exploit variation in treatment intensity across counties stemming from differences in the income gains after the switch from the regular to the expanded CTC." The authors use a county's poverty rate to proxy these CTC gains, and operationalize poverty status as a categorical variable (low, medium, high poverty) in their main models. Results are then presented as effects of the CTC expansion on consumption (visits and spending) among high- and medium- poverty counties relative to low-poverty counties. Estimates for each establishment type are shown in Figures 1 and 2, with overlapping confidence intervals. Point estimates for high- and medium- poverty counties are also quite similar, with sometimes larger point estimates for medium-poverty counties. The manuscript would be strengthened if the authors briefly discussed this pattern of results (given that there appears to be little difference between the high- and moderate- "treatment" intensity

groups).

5) The authors include a somewhat limited set of covariates in their model and do not control for other policy changes that occurred over this study period (of which there were many). This manuscript would be strengthened by further justification that these results are not due to other policy changes.

Reviewer #3:

Remarks to the Author:

Title: Consumption Responses to an Unconditional Child Allowance in the United States

In this paper, the authors investigate how changes to the Child Tax Credit (CTC) in tax year 2021 affected expenditure patterns and in-person visits. Their identification strategy relies on comparing cell phone and credit/debit card information across counties categorized as low-, medium-, and high-poverty, assuming higher poverty areas were more treated by the policy change than lower poverty areas. The authors present an extensive set of results based on numerous sample restrictions and specifications. The authors conclude that the expanded CTC resulted in more visits to child care facilities and increased spending at restaurants and grocery stores.

I enjoyed reading this paper and commend the authors on their thorough and careful analysis. This is an important topic and an exciting area of research. I've compiled a list of comments below that I hope will help improve the paper.

1. Big picture issues

a. It is important to spend more time on the mechanisms by which the CTC would lead to higher child care visitations. Can you show that parents were more likely to go back to work in person? It seems like you are implying that the CTC allowed parents to afford child care that they were foregoing previously, the cost of which prevent employment. If not, we are left wondering why more cash on hand would lead to higher child care visits. The discussion section touches on these points but raises more questions than it answers.

b. A major limitation of the paper is that you do not have expenditure data on child care services. I understand that this is a data issue, but this should be highlighted in both the intro and conclusion.

i. Another major limitation is that you do not have spending data for 2019, meaning that you are effectively using 2020 as the baseline for any deviations in spending in 2021 and beyond. This seems very problematic.

c. What is the economic significance of the results? Why should we care if a \$1000 increase in annual income resulted in 6.7 percent more child care centers?

d. A better job should be done to synthesize the different, and sometimes contradictory, results. As it currently stands, the authors present many results, most of which are insignificant or inconclusive, and do their best to weave a narrative between them. It leaves the reader wondering how much confidence should be placed in the findings.

2. Identification strategy and methodology

a. The baseline specification must include state fixed effects. In fact, some might say that state-by-month fixed effects would be more appropriate, though I think that such a specification would be suitable for a robustness check.

b. State-level policies that affected local economic activity were in flux during the period of analysis.

Lockdowns, school policy, mask mandates, unemployment insurance, and so on. All standard errors should be clustered at the state level.

c. I had to read the description of the main specification multiple times. Specifically, saying that you have an indicator of the poverty rate is confusing. It would be better to say an indicator of the poverty rate tercile, or something similar.

d. The most convincing results, by far, were the even study plots and should be presented as the main results. Not only do these charts help support the parallel trends assumption, but they also avoid the awkward (and somewhat arbitrary) monthly groupings.

i. Specifically, the option to take monthly checks or the lump sum payment makes identification trickier than it would be otherwise. Plus, refunds could have been received before March 2022. Avoid all of this by simply using the event study specification which doesn't assume that certain months are more/less treated. Let the data show you dynamics.

ii. If you want a nice simple DD estimate, perhaps you could end the panel in Dec 2021 and just use the average estimate for the second half of 2021.

iii. Perhaps I missed it: why does Figure F1 not start before January 2021? The spending data is limited to 2021 and beyond but not the cell phone data, correct?

e. A more recent poverty rate should be used as a robustness check. Why not use 2018 ACS data to define the poverty terciles? 2010 data from Op. Insights seems like an odd choice.

f. More should be said about the variation in COVID policies that could be contemporaneous with the monthly CTC checks. If low-poverty areas, which are concentrated in the south (Figure G1), were treated to different COVID policies in the second half of 2021 this would bias the estimates.

State-by-month fixed effects would be the most straightforward way to address this concern and would avoid time-intensive data collection.

3. Other assorted comments

a. The first paragraph of the introduction feels very out of place and disrupts the flow of the entire paper.

b. My prior is that lower-poverty areas have less remote-friendly work options. Can this be discussed as it relates to child care? This could be relevant for return-to-work considerations that occurred around the time that the CTC was expanded.

c. It would be wise to downplay or cut the "cover more than 328 million U.S. residents (99.6 percent of the population)" comment. It oversells the data you actually use for the analysis, which is likely not representative.

d. I liked many of the robustness checks, especially the school falsification check. Can you do the same with public parks? Other places that families might frequent?

e. Even though I prefer the simple figures, the full regression tables should be included in the appendix.

f. More attention should be given to the potential labor supply effects, and therefore family income consequences, of an unconditional, refundable CTC (see Han, Meyer, and Sullivan, 2022).

g. I would recommend reorganizing the paper to demonstrate the validity of the parallel trends assumption before you get into the main results. Again, this can all be accomplished by focusing on the event study results.

h. An appendix section detailing the visitation data should be included. Most readers will not be intimately familiar with these data.

i. I am not sure I fully agree that the results are understated due to differential cell phone use. Couldn't families with cell phones also have a differential response to the CTC?

j. Unemployment data at the county level is available from the BLS and should be included as controls. Or at least justify the use of Op. Insights data.

Author Rebuttal to Initial comments

Comments from Reviewer #1:

Interesting and important question with great data. I appreciate going beyond the standard CES or Pulse data.

1.1) Are you accounting for EITC refunds also happening in March-May?

Response: We appreciate the reviewer’s support and inquiries. Regarding the EITC: given that our mobility estimates are set relative to mobility patterns in the same month in 2019, our baseline estimates effectively take into account how policies that existed prior to 2021, such as the EITC, affect consumption patterns. The one difference in 2021 relative to 2019 could be the small adjustments to EITC benefit levels for childless adults as part of the American Rescue Plan Act in 2021 (which affected the size of some tax returns in spring 2022). In our revised analysis, we find that the potential impact of the childless EITC expansion on differential consumption patterns is likely to be very small relative to the expanded CTC. In Appendix B (and copied below), we show that the average household in high-poverty counties only experiences a \$20 absolute income gain relative to the average household in low-poverty counties; this absolute income gain is around 6 percent of the gains that we measure for the expanded CTC (see Appendix Tables B1-B3). It is thus unlikely that the childless EITC expansion meaningfully and differentially affects consumption patterns across high-, medium-, and low-mobility counties. Given that we cannot seasonally adjust our credit card spending data (data for 2019 are not available), differential receipt of the EITC across high-, medium, and low-poverty counties could slightly affect our estimates of consumption on the intensive margin.

Table: Association of county poverty status and gains from the childless EITC expansion

	Absolute \$ Gain due to Childless EITC Expansion	Log \$ Gain due to Childless EITC Expansion	Absolute \$ Gain due to Childless EITC Expansion	Log \$ Gain due to Childless EITC Expansion
Medium Poverty County X Post-Expansion	14.97*** (2.637)	0.001*** (0.000)		
High Poverty County X	20.28***	0.002***		

Post-Expansion	(3.576)	(0.000)	
Country Poverty Rate X			184.83***
Post-Expansion			0.016***
			(22.534)
			(0.002)

Note: This analysis is limited to counties observed in the CPS-ASEC. "Absolute Gain" refers to the additional dollar value of income due to the childless EITC expansion.

1.2) Many purchases were made online and not in person in 2021. If anything, I'd expect more purchases online for higher income people, which could drive up the relative number of in-person purchases that you'll attribute to the CTC. Is this a threat to identification?

Response: We appreciate the opportunity to clarify this point. In Footnote 5, we note that: For around 60% of establishments, we have data on mean debit/credit card spending for transactions that occur online. Given the imperfect coverage and generally low values of online spending, we do not include these in our primary analysis. In additional analyses, however, we find positive but highly imprecisely estimated differences in online spending amounts for high- and medium-poverty counties relative to low-poverty counties after the CTC expansion relative to before at any of our establishment types. Put differently, we do not find evidence that differential online spending across counties before versus after the CTC reform should bias our findings upwards.

1.3) How do your results compare with "Spending Responses to High-Frequency Shifts in Payment Timing: Evidence from the Earned Income Tax Credit" by Aladangady et al? Be sure to cite this paper along with Goodman-Bacon & McGranahan, 2008.

Response: Aladangady et al. (2018; AEJ forthcoming) explore how shifts in the timing of EITC payments to households affect spending. Using high-frequency spending data coupled with variation in the timing of tax refunds stemming from the 2017 PATH Act, they document considerable consumption responses to a two-week shift in the timing of payments. Their results provide clear evidence that spending out of EITC refunds is not smoothed evenly throughout the year. Evidence from our event study analysis of reduced spending in restaurants and grocery stores in partially treated months also suggests that spending out of CTC refunds is not smoothed evenly throughout the year, even though – compared to Aladangady et al. (2018; AEJ forthcoming) – our results combine the effect of a shift in timing and an increase in the level of the transfer. We acknowledge this link to their paper in the Discussion section.

1.4) Paragraph on page 15 is too long and dense.

Response: We appreciate the reviewer pointing this out and we have now revised the paragraph accordingly.

Comments from Reviewer #2:

The study “Consumption Responses to an Unconditional Child Allowance in the United States” examines consumption responses to the 2021 Child Tax Credit expansion using anonymized mobile-location data and debit/credit card data. This is an innovative study design and the authors well motivate the study’s aims. I have several suggestions that I hope might improve the manuscript.

2.1) The manuscript would be improved if the authors more thoroughly introduced and justified some of their methodological decisions. In particular, the authors should explain why they chose March-May 2022 as “treatment months” for analyzing lump-sum CTC effects on consumption. While issuance timing of the monthly 2021 CTC payments around the 15th of each month from July to December 2021 is clear, there is less certainty about the issuance and receipt timing of lump-sum CTC. Similarly, the authors should also discuss measurement error in their proxy for lump-sum CTC “treatment” (represented by an indicator variable for “lump-sum treatment” months) given that there is considerable variability in the timing and magnitude of lump-sum tax credit receipt, which goes uncaptured by an indicator variable. Several studies present data showing that the timing of lump-sum tax credit receipt is far from uniform around tax time (Jones & Michelmore; Aladgady et al., 2018) and is correlated with household income (Maag, Roll, Oliphant, 2016). The authors should consider how this source of measurement error in their treatment variable could influence their results. (This issue is less concerning for treatment T, a binary indicator of the timing of monthly CTC receipt, since the monthly CTC payments were received around the 15th of each month, and thus, were fairly uniform in terms of both timing and total monthly outlays over the 6-month period).

Response: We appreciate the reviewer’s feedback and recommendations for improvement. In the revised manuscript, we now give greater discussion to timing of the lump-sum payments. Specifically, we cite IRS evidence of the distribution of refundable tax credit payments, noting that around half of EITC benefits (a refundable tax credit for which we have available data on payment timing) are paid in the final week of February, while approximately three-fourths are paid out by the end of March, and approximately 90 percent are paid out by the end of April. These estimates are from the Internal Revenue Service: Research, Applied Analytics, & Statistics (as cited in Aladgady et al., 2018). We do not have direct data on the timing or location of the lump-sum CTC distributions in 2022; as such, the reviewer is right that identifying the lump-sum payment timing is far more challenging than isolating the monthly payments. In our primary models, we identify the lump-sum treatment period of March and April 2022. In our revised manuscript, we now also include sensitivity tests in which we identify the lump-sum period as February, March, and April 2022 (results are comparable). Moreover, we have moved up our event study figures to the main text; the event study plots show effect estimates by month and do not force us to directly identify the lump-sum treatment timing. In all cases, we emphasize more to the reader that there is less certainty around issuance and receipt timing for the lump-sum CTC payments.

2.2) The authors do not include county fixed-effects in their regression models, which seem appropriate in order to control for any time-invariant, unmeasured characteristics of the county. (I am especially concerned about unobserved factors that are correlated with poverty status, which the authors use as a proxy for CTC gains.) Can the authors describe why county fixed-effects were omitted from their models? Ideally, the authors could show that their results are robust to the inclusion of county fixed-effects.

Response: In our revised manuscript, we now present findings from several alternative specifications that the reviewers have recommended: (1) our baseline estimates with year/month fixed effects and a new set of controls for county unemployment trends from the Bureau of Labor Statistics, state-month policies on COVID mobility restrictions, state-month availability of SNAP Emergency Allotments, and state-month availability of expanded unemployment benefits; (2) our baseline estimates but also with state fixed effects; (3) our baseline estimates but also with interacted state-year/month fixed effects; (4) our baseline estimates but also with county fixed effects; (5) a model in which we create poverty bins based on the state-specific poverty distribution. Our primary findings regarding increased extensive-margin consumption at childcare centers hold across all five models. That said, our baseline model (the first of the five listed above) more consistently passes our placebo tests and pre-trends analyses and thus remains our preferred specification, as we now clarify in the manuscript. We now show results across each of these models, however, to emphasize the consistency of our primary findings across each. The table below illustrates how we demonstrate this point in the revised manuscript.

Table: Estimated effect of CTC expansion on visits to child care centers by model specification

	Child Care: Pre-Trend (Jan. – June 2020)	Child Care: All Payments	Child Care: Monthly Payments	Child Care: Lump-Sum Payments
Month FE				
Medium Poverty X Treatment	-0.004 (0.002)	0.030*** (0.008)	0.024** (0.008)	0.037** (0.011)
High Poverty X Treatment	0.005 (0.003)	0.029** (0.009)	0.024* (0.011)	0.036** (0.012)
State FE, Month FE				
Medium Poverty X Treatment	-0.003 (0.002)	0.032*** (0.008)	0.029** (0.009)	0.040*** (0.011)

High Poverty X Treatment	0.007* (0.003)	0.032** (0.010)	0.029* (0.012)	0.043*** (0.012)
				
State x Month FE				
Medium Poverty X Treatment	-0.005* (0.002)	0.027** (0.008)	0.023** (0.008)	0.032** (0.010)
High Poverty X Treatment	0.002 (0.003)	0.031** (0.011)	0.027* (0.012)	0.035** (0.012)
				
County FE, Month FE				
Medium Poverty X Treatment	-0.002 (0.002)	0.037*** (0.008)	0.036*** (0.009)	0.048*** (0.010)
High Poverty X Treatment	0.010** (0.003)	0.043*** (0.011)	0.040** (0.012)	0.061*** (0.012)
				
Within-State Poverty Bins (State, Month FE)				
Medium Poverty X Treatment	0.002 (0.003)	0.022** (0.007)	0.023** (0.007)	0.021* (0.009)
High Poverty X Treatment	0.002 (0.002)	0.025** (0.008)	0.027** (0.009)	0.019 (0.010)

Standard errors in parentheses. * $p < 0.05$, ** $p < 0.01$, *** $p < 0.001$

2.3) For identification, the authors “exploit variation in treatment intensity across counties stemming from differences in the income gains after the switch from the regular to the expanded CTC.” The authors use a county’s poverty rate to proxy these CTC gains, and operationalize poverty status as a categorical variable (low, medium, high poverty) in their main models. Results are then presented as effects of the CTC expansion on consumption (visits and spending) among high- and medium- poverty counties relative to low-poverty counties. Estimates for each establishment type are shown in Figures 1 and 2, with overlapping confidence intervals. Point estimates for high- and medium- poverty counties are also quite similar, with sometimes larger point estimates for medium-poverty counties. The manuscript would be strengthened if the authors briefly discussed this pattern of results (given that there appears to be little difference between the high- and moderate- “treatment” intensity groups).

Response: The author is correct that, for many consumption categories, point estimates are similar for medium- and high-poverty counties relative to low-poverty counties. We now discuss in the text that these patterns suggest that the proportional effect of the CTC on consumption may not increase linearly at higher levels of CTC exposure. This further justifies the use of the poverty bins rather than a continuous measure of the poverty rate (though we present these results also in the appendix). For our primary outcome of interest, the findings suggest that increases in visits to childcare centers were comparable for families in medium- and high-poverty counties relative to low-poverty counties.

2.4) The authors include a somewhat limited set of covariates in their model and do not control for other policy changes that occurred over this study period (of which there were many). This manuscript would be strengthened by further justification that these results are not due to other policy changes.

Response: We have now altered our estimation strategy to include state-month variation in policy availability that could otherwise bias our estimates. Specifically, our revised models now include state-month indicators of whether the state provides SNAP Emergency Allotments (which allows states to increase SNAP benefit levels to the maximum benefit amount), and state-month indicators on availability of expanded unemployment benefits (which some states began withdrawing in the summer of 2021). Additionally, we have added in county-month data on unemployment trends from the Bureau of Labor Statistics, and state-month policies on COVID mobility restrictions from the Oxford COVID-19 Government Response Tracker. These additional covariates help to further account for policy changes that have occurred throughout the study period that may have differentially affected consumption patterns across low-, medium-, and high-poverty counties. Moreover, please see Response 1.1 for a discussion of how we treat the American Rescue Plan's childless EITC expansion.

Comments from Reviewer #3:

In this paper, the authors investigate how changes to the Child Tax Credit (CTC) in tax year 2021 affected expenditure patterns and in-person visits. Their identification strategy relies on comparing cell phone and credit/debit card information across counties categorized as low-, medium-, and high-poverty, assuming higher poverty areas were more treated by the policy change than lower poverty areas. The authors present an extensive set of results based on numerous sample restrictions and specifications. The authors conclude that the expanded CTC resulted in more visits to child care facilities and increased spending at restaurants and grocery stores.

I enjoyed reading this paper and commend the authors on their thorough and careful analysis. This is an important topic and an exciting area of research. I've compiled a list of comments below that I hope will help improve the paper.

3.1. Big picture issues:

a. It is important to spend more time on the mechanisms by which the CTC would lead to higher child care visitations. Can you show that parents were more likely to go back to work in person? It seems like you are implying that the CTC allowed parents to afford child care that they were foregoing previously, the cost of which prevent employment. If not, we are left wondering why more cash on hand would lead to higher child care visits. The discussion section touches on these points but raises more questions than it answers.

Response: Our findings suggest that the CTC contributed to increases in visits to formal child care centers. Households may now afford formal childcare, as either a complement or a substitute to informal childcare. Though an analysis of labor supply decisions is beyond the scope of this study, it is possible that these findings could contribute to explaining the expanded CTC's lack of negative employment effects in 2021 (Ananat et al., 2022; Pilkauskas et al., 2022; Enriquez et al., 2023). First, increases in childcare consumption are not incompatible with the lack of employment effects. Gennetian et al. (2022, 2023) provide evidence that child-related cash transfers lead to increased expenditure in childcare (17%), with no associated employment effects. Second, it is possible that the unconditional cash transfer might have lowered participation or hour constraints for some parents, whilst reducing employment incentives for others. Our findings from the Annual Social and Economic Supplement (ASEC) of the Current Population Survey (CPS) that, following the CTC expansion, single parents with children living in poorer counties became less likely to report being non-employed or working part-time due to child care problems, suggest that the first channel might be at play, at least for part of the population. Finally, the temporary nature of the expanded CTC might have reduced individual incentives to find a stable job in response to increased cash assistance. Unfortunately, our data do not allow us to uncover the precise mechanisms behind the effects that we document, but these do not seem to be inconsistent with the lack of employment effects. We now discuss these points in the Discussion section.

3.2. A major limitation of the paper is that you do not have expenditure data on child care services. I understand that this is a data issue, but this should be highlighted in both the intro and conclusion.

Response: We acknowledge in the Data and Findings sections that we measure visits to child care centers, but do not have credit/debit card spending data for child care centers. We also more carefully represent our results in the Discussion section. Since we do not summarize our findings in the Introduction, we have not added this information there.

3.3. Another major limitation is that you do not have spending data for 2019, meaning that you are effectively using 2020 as the baseline for any deviations in spending in 2021 and beyond. This seems very problematic.

Response: We acknowledge in the paper that we are not able to seasonally adjust our spending data in the way that we can for our mobility data. We do not use any spending data from 2020 in our analysis (we do not seasonally adjust to a 2020 baseline) given that spending patterns for

2020 are unlikely to be a reliable baseline. The precision of our spending estimates thus relies on an assumption that seasonal variation in spending will apply similarly to medium- and-high poverty counties relative to low-poverty counties. We now acknowledge this in our Data section.

3.4. What is the economic significance of the results? Why should we care if a \$1000 increase in annual income resulted in 6.7 percent more child care centers?

Response: Thank you for raising this point. Rescaling our reduced-form estimates of the effect of the expended CTC on childcare visits by the size of the income shock allows for simpler benchmarking of our estimates with those of other similar programs. An informative comparison is with estimates of the effect of the Child and Dependent Care Credit (CDCC) on paid childcare participation (Pepin, 2020). The CDCC is a nonrefundable tax credit based on taxpayers' income and childcare expenses, aimed at reducing childcare costs. Using cross-state and cross-family-size variation in the generosity of the CDCC induced by a 2003 CDCC expansion, Pepin documents that a 10 percent increase in CDCC benefits increases annual paid child care participation by 4–5 percent (elasticity of 0.5) among households with children younger than 13 years old. Our results imply an elasticity of 0.9, indicating larger responses to the expanded CTC. Two factors are likely to explain the larger elasticity in our context: (i) in-person visits capture both participation and intensive-margin responses; (ii) responses to the CTC may be larger due to its refundable nature, which implies that low-income families are also largely treated. We now clarify this point in the Findings section.

3.5. A better job should be done to synthesize the different, and sometimes contradictory, results. As it currently stands, the authors present many results, most of which are insignificant or inconclusive, and do their best to weave a narrative between them. It leaves the reader wondering how much confidence should be placed in the findings.

Response: We have revised the text to describe our results in a synthetic, yet precise manner in the manuscript, giving more prominence to our primary result of in-person visits to child care centers.

3.6. Identification strategy and methodology: The baseline specification must include state fixed effects. In fact, some might say that state-by-month fixed effects would be more appropriate, though I think that such a specification would be suitable for a robustness check.

Response: In our revised manuscript, we now (1) test the sensitivity of our results across a range of alternative model specifications, including those that the reviewer suggests and (2) give greater clarity to why we prefer our baseline estimates with year/month fixed effects (plus a new set of controls for county unemployment trends from the Bureau of Labor Statistics, state-month policies on COVID mobility restrictions, state-month availability of SNAP Emergency Allotments, and state-month availability of expanded unemployment benefits).

Regarding the first point, Appendix G (see also the Table provided in response to Reviewer 2) now validates the consistency of our primary findings across several different model specifications: our baseline model, a model with state FE, a model with state by year-month FE, a model with county FE, and a model that creates poverty bins based on the state-specific poverty distribution.

Regarding the second point, we elaborate on why we prefer our baseline model that excludes state fixed effects. The primary challenge of including state fixed effects is that states vary widely in their balance of low-, medium-, and high-poverty counties. When creating poverty bins based on the poverty rate distribution of all counties, Mississippi, for example, has only 3 low-poverty counties but 67 high-poverty counties. Rhode Island, conversely, has no high-poverty counties. Simply including state fixed effects effectively forces our model to compare most of Mississippi's counties to a reference group of merely three counties, and offers no full comparison in states such as Rhode Island. A stronger approach to avoid losing information from each state is to create state-specific poverty terciles across each state's counties, as presented in the fifth row of Tables G3 to G5. This approach ensures that one-third of Mississippi's counties are identified as each of low-, medium-, high-poverty counties, avoiding the small-reference-group challenge documented above. The conceptual challenge that remains is that, using this approach, relatively *low*-poverty counties in Mississippi (to continue the example) have a mean poverty rate of 17.9%, which is higher than the mean poverty rate of 12% in relatively *high*-poverty counties in Connecticut. If the purpose of our model is to apply a treatment that aligns well with gains from the 2021 CTC expansion, then this within-state approach objectively performs worse than our baseline model. Given that the more restrictive model with state fixed effects (or state by year-month fixed effects) more strongly excludes omitted variable bias, however, we also corroborate our primary findings with results from these specifications in Appendix G. Our primary findings from the baseline models of increased childcare consumption holds across all model specifications evaluated.

3.7. State-level policies that affected local economic activity were in flux during the period of analysis. Lockdowns, school policy, mask mandates, unemployment insurance, and so on. All standard errors should be clustered at the state level.

Response: All of our estimates (in the original and revised manuscripts) cluster standard errors at the state level, which we now acknowledge in the revised text. Moreover, we now also include (1) state-month indicators of stringency of COVID containment and health measures from the Oxford COVID-19 Government Response Tracker, and (2) state-month indicators of availability of SNAP Emergency Allotments and expanded unemployment benefits.

3.8. I had to read the description of the main specification multiple times. Specifically, saying that you have an indicator of the poverty rate is confusing. It would be better to say an indicator of the poverty rate tercile, or something similar.

Response: We agree with the reviewer and we now clarify that we segment counties into poverty terciles or poverty bins.

3.9. The most convincing results, by far, were the event study plots and should be presented as the main results. Not only do these charts help support the parallel trends assumption, but they also avoid the awkward (and somewhat arbitrary) monthly groupings.

Response: We agree with the reviewer's suggestion to bring the event study plots into the main analysis. We now give greater discussion to these figures in the Findings section.

3.10. Specifically, the option to take monthly checks or the lump sum payment makes identification trickier than it would be otherwise. Plus, refunds could have been received before March 2022. Avoid all of this by simply using the event study specification which doesn't assume that certain months are more/less treated. Let the data show you dynamics.

Response: We agree with the reviewer's suggestion to bring the event study plots into the main analysis. We also include alternative operationalizations of our lump-sum treatment effects (see response to reviewer comments 2.1 and 2.2).

3.11. If you want a nice simple DD estimate, perhaps you could end the panel in Dec 2021 and just use the average estimate for the second half of 2021.

Response: We now clarify in the manuscript that our estimates for "Monthly payments" (as visualized in Figure 1, for example) provide this exact estimate.

3.12. Perhaps I missed it: why does Figure F1 not start before January 2021? The spending data is limited to 2021 and beyond but not the cell phone data, correct?

Response: Per the reviewer's request, we have included a version of the event study plot for our mobility estimates that begins in July 2020 (one year before the start of the monthly CTC payments). We prioritize our analyses that use January to June 2021 as the pre-treatment reference period, as the second half of 2020 was more fraught with COVID cases and deaths, continued school closures, the expiration of the Pandemic Unemployment Compensation, and more. Thus, in our primary estimates and the event study we include in the main part of the manuscript, we continue with the presentation of results that starts in January 2021. In the Appendix, we have included the figure copied below, which begins the time series at July 2020 instead. The figure demonstrates that our primary finding regarding increased visits to child care centers after the CTC payments still holds when extending the pre-treatment period.

Figure: Extended event study estimates of the effect of expanded CTC on in-person visits to different establishment types in high-poverty counties relative to low-poverty counties (pre-treatment period beginning in July 2020)

3.13. A more recent poverty rate should be used as a robustness check. Why not use 2018 ACS data to define the poverty terciles? 2010 data from Op. Insights seems like an odd choice.

Response: We agree with the reviewer’s recommendation and appreciate the suggestion. We have now replaced our poverty indicators with 5-year averages of ACS county poverty rates spanning 2014-2018. We avoid any data from 2019-2021 given biased response rates and potentially-altered poverty rates due to the COVID-19 pandemic. The figure below, provided for the reviewers, demonstrates that our 2010 and 2018 poverty rates are strongly correlated, which is why our results do not meaningfully differ after updating the poverty indicators.

Figure: Comparison of county poverty rates: 2014-2018 estimates (new measure used in the paper) versus 2010 estimates (measure used in prior draft)

3.14. More should be said about the variation in COVID policies that could be contemporaneous with the monthly CTC checks. If low-poverty areas, which are concentrated in the south (Figure G1), were treated to different COVID policies in the second half of 2021 this would bias the estimates. State-by-month fixed effects would be the most straightforward way to address this concern and would avoid time-intensive data collection.

Response: The reviewer raises an important point regarding state-month variation in COVID restrictions, which we now address in two ways. First, in all of our estimates, we now include data from the Oxford COVID-19 Government Response Tracker on state-month COVID restrictions. Specifically, we include the indicators of *stringency* (the strictness of ‘lockdown style’ policies that primarily restrict people’s behavior) and *containment and health measures* (capturing ‘lockdown’ restrictions and closures with measures such as testing policy and contact tracing, short term investment in healthcare, as well investments in vaccines). Second, we also include estimates that include state-by-year/month fixed effects, as discussed above. Our primary findings hold when including these interacted state-month fixed effects.

3.15. The first paragraph of the introduction feels very out of place and disrupts the flow of the entire paper.

Response: We have now moved this paragraph down per the reviewer’s request.

3.16. My prior is that lower-poverty areas have less remote-friendly work options. Can this be discussed as it relates to child care? This could be relevant for return-to-work considerations that occurred around the time that the CTC was expanded.

Response: We agree with the reviewer’s suggestion to discuss remote work in relation to our child care findings. Evidence from the Current Population Survey demonstrates that employed adults in higher-poverty counties are less likely to have the option to work remotely relative to employed adults in lower-poverty counties. The binned scatterplot below demonstrates this point, though only among the small subset of counties (n=280) identifiable in the monthly CPS files. Specifically, the scatterplot shows a negative correlation ($r = -0.24$) of the county’s poverty rate (5-year average of 2014-2018) and the share of employed adults with the ability to work remotely (average over January to June 2021). We mention this relationship in our Discussion section. Specifically, we note the likelihood that workers in higher poverty counties are less able to have the option to work remotely and, thus, more likely to need child care support in order to continue working. It is possible that this fact contributes to the larger increase in child care consumption for higher poverty counties after the introduction of the CTC, though we cannot directly evaluate this given the small sample of counties available in the Current Population Survey data (the dataset in which we can estimate remote work shares).

Figure: County poverty rate by remote work possibility among employed adults

Note: County poverty rates are 5-year averages from the American Community Survey. The share of employed adults able to work remotely is from the Current Population Survey (CPS) Basic Monthly files (January to June 2021). The sample is limited to the 280 identifiable counties in the CPS files.

3.17. It would be wise to downplay or cut the “cover more than 328 million U.S. residents (99.6 percent of the population)” comment. It oversells the data you actually use for the analysis, which is likely not representative.

Response: We have now removed this wording to avoid overselling the coverage of our data.

3.18. I liked many of the robustness checks, especially the school falsification check. Can you do the same with public parks? Other places that families might frequent?

Response: We have explored the possibility of adding falsification tests beyond schools. The closest option to the reviewer’s recommendation is NAICS code 712190, or “Nature Parks and Other Similar Institutions.” However, this category is not restricted to free, public parks; instead, it also includes attractions with entrance fees, such as bird or wildlife sanctuaries, natural wonder tourist attractions (e.g., caverns, waterfalls), conservation areas, nature centers or preserves, and national parks. Therefore, it does not meet the same standards as public schools for use in a falsification test.

3.19. Even though I prefer the simple figures, the full regression tables should be included in the appendix.

Response: We now include the regression tables in Appendix G.

3.20. More attention should be given to the potential labor supply effects, and therefore family income consequences, of an unconditional, refundable CTC (see Han, Meyer, and Sullivan, 2022).

Response: We now include a discussion of the existing evidence on labor supply effects in the Discussion section.

3.21. I would recommend reorganizing the paper to demonstrate the validity of the parallel trends assumption before you get into the main results. Again, this can all be accomplished by focusing on the event study results.

Response: We now illustrate our event-study results in our primary Findings section and highlight that they provide reassuring evidence of the validity of the parallel trends assumption. We opted for leaving the discussion of measurement validation, identification tests and robustness checks after the illustration of the main findings, since we felt this would make the manuscript easier to follow.

3.22. An appendix section detailing the visitation data should be included. Most readers will not be intimately familiar with these data.

Response: We have now added an appendix that provides further details on the SafeGraph data (see Appendix C).

3.23. I am not sure I fully agree that the results are understated due to differential cell phone use. Couldn’t families with cell phones also have a differential response to the CTC?

Response: Thank you for raising this point. Our statement assumes that families with and without smartphones have similar preferences for consumption (of childcare and other items). If families without smartphones have different consumption preferences (e.g. are averse to the use of childcare), then we would be overestimating the effect of interest. We better qualified our statement in the manuscript.

3.24. Unemployment data at the county level is available from the BLS and should be included as controls. Or at least justify the use of Op. Insights data.

Response: We have followed the reviewer's suggestion to use Bureau of Labor Statistics county-level unemployment data in all estimates, rather than the Opportunity Insights data used before. This gives us full county-month coverage, and we now include the unemployment data as a control in our baseline models. We again thank the reviewer for the useful set of recommendations.

Decision Letter, first revision:

2nd June 2023

Dear Dr. Parolin,

RE: "Consumption Responses to an Unconditional Child Allowance in the United States"

Thank you for submitting your revised manuscript and for all your work on the revision.

Although your manuscript has been revised in response to reviewer comments, it does not fully comply with our editorial policies and formatting requirements. In particular, your manuscript does not currently follow our required order for sections, and statistical results are not fully reported according to our requirements. In order to ensure that reviewers have the opportunity to evaluate a version of your manuscript that is close to its final form, we ask that you revise to address these points:

1) Please ensure that all inferential statistical results discussed in the main text are fully reported, including coefficients, p-values, and confidence intervals. If you would prefer to refer to a table rather than reporting all results fully in-text, please ensure that the table includes exact p-values (for all $p > 0.001$) and confidence intervals. Asterisks should not be used to mark statistical significance as a substitute for reporting p-values.

2) Please revise to follow our section order: Introduction (no subheadings permitted), Results (subheadings permitted), Discussion (subheadings permitted), Methods (subheadings permitted). Although I realize that that is order is non-standard in the social sciences, we are unfortunately not able to alter it. We recommend that you provide enough information about your study design at the end of the Introduction or start of the Results section to allow readers to follow your discussion of results. Further details can then be left for the Methods section.

In addition to addressing these core requirements, we also strongly encourage you to make an

additional change now, in order to save time at a later stage. Our format does not permit footnotes, and these should be incorporated into the text or removed. Although we would not insist on this change prior to further peer review, we do encourage you to implement it as you complete the necessary statistical/formatting revisions.

Before we can send the manuscript back to our reviewers, we ask that you revise it to ensure that it complies with our policies and addresses these core concerns. To assist with this process, I have attached another copy of our checklist. If you are uncertain as to how to address any of the points in the checklist, please don't hesitate to contact me.

[REDACTED]

Thank you in advance for attending to these requests and I look forward to receiving your revised manuscript.

Sincerely,
Aisha

Aisha Bradshaw, PhD
Senior Editor
Nature Human Behaviour

Author Rebuttal, first revision:

Response to Reviewers, “Consumption Responses to an Unconditional Child Allowance in the United States” at Nature Human Behaviour

We greatly appreciate the useful set of suggestions from the editors and three reviewers. In this response letter, we first outline the broad changes made to our revised manuscript while responding to the requests of the editorial team. We then proceed to a point-by-point reply to the reviewers' comments.

Comments from Editorial Team:

1) Address reviewer concerns about your regression framework, including fixed effects and clustered standard errors as recommended by the reviewers.

Response: In our revised manuscript, we now present findings from several alternative specifications that the reviewers have recommended: (1) our baseline estimates with year/month fixed effects and a new set of controls for county unemployment trends from the Bureau of Labor

Statistics, state-month policies on COVID mobility restrictions, state-month availability of SNAP Emergency Allotments, and state-month availability of expanded unemployment benefits; (2) our baseline estimates but also with state fixed effects; (3) our baseline estimates but also with interacted state-year/month fixed effects; (4) our baseline estimates but also with county fixed effects; and (5) a model in which we create poverty bins based on the state-specific poverty distribution among counties. Our primary findings regarding increased extensive-margin consumption at childcare centers hold across all models. That said, our baseline model (the first of the five listed above) more consistently passes our placebo tests and pre-trends analyses and thus remains our preferred specification. We elaborate on these points below. We now show results across all five models, however, to emphasize the consistency of our primary findings across each. We also now clarify that all of our models include standard errors clustered at the state level.

2) Address reviewer concerns regarding the comprehensiveness of the covariate set included and the potential for a number of other variables to confound the estimates.

Response: We have now altered our estimation strategy to expand the set of covariates in our models, in line with reviewer suggestions. Specifically, our revised models now include state-month indicators of whether the state provides SNAP Emergency Allotments (which allows states to increase SNAP benefit levels to the maximum benefit amount), and state-month indicators on availability of expanded unemployment benefits (which some states began withdrawing in the summer of 2021). Additionally, we have added in county-month data on unemployment trends from the Bureau of Labor Statistics, and state-month policies on COVID mobility restrictions from the Oxford COVID-19 Government Response Tracker. These additional covariates help to further account for policy changes that have occurred throughout the study period that may have differentially affected consumption patterns across low-, medium-, and high-poverty counties.

3) Ensure that your methodological choices are clear and fully justified, and that the pattern of results is clearly explained.

Response: We have now elaborated on our methodological choices in the manuscript, and we have done more to clarify the patterns of our findings. On our methodological choices, we now show results across different specifications and discuss the advantages and limitations of all in our context. We streamlined the illustration of our findings, refocusing the discussion on our primary findings and highlighting any deviations from those. We also do more to benchmark and contextualize our child care findings, comparing them to existing estimates of analogous programs, where we devote particular attention to emphasizing institutional differences with the expanded CTC.

4) Include a thorough discussion of the limitations of the data and analyses, and transparently discuss the implications of these.

Response: We now include a broader discussion of our study's limitations, as well as their implications. For instance, in the Data section, we clarify the advantages and limitations of our data. We explain that we can deseasonalize only the in-person visits data and clarify the assumptions we rely upon for the interpretation of our spending data, which – due to data availability limitations – cannot be deseasonalized. In the Identification Strategy section, we acknowledge uncertainty of precise timing of lumpsum payments and then assess the robustness of our findings to different assumptions about payment timing in the Robustness Checks section. We discuss issues related to the coverage and representativeness of our data in the Measurement Validation section, and devote particular attention to specifying what data our results rely upon – in-person visits vs credit/debit card spending – throughout the manuscript.

Response: In our revision, we made sure to comply fully with the journal's editorial policies and formatting requirements. In particular, we devoted special attention to using appropriate language when describing our results. While our primary results are not null, we made sure to appropriately characterize all results in the text, including null results.

Additionally, we note to reviewers that the revised formatting of the manuscript (with Results presented before Methods, and with no footnotes) is due to journal guidelines and cannot be altered.

Comments from Reviewer #1:

Interesting and important question with great data. I appreciate going beyond the standard CES or Pulse data.

1.1) Are you accounting for EITC refunds also happening in March-May?

Response: We appreciate the reviewer's support and inquiries. Regarding the EITC: given that our mobility estimates are set relative to mobility patterns in the same month in 2019, our baseline estimates effectively take into account how policies that existed prior to 2021, such as the EITC, affect consumption patterns. The one difference in 2021 relative to 2019 could be the small adjustments to EITC benefit levels for childless adults as part of the American Rescue Plan Act in 2021 (which affected the size of some tax returns in spring 2022). In our revised analysis, we find that the potential impact of the childless EITC expansion on differential consumption patterns is likely to be very small relative to the expanded CTC. In Appendix B (and copied below), we show that the average household in high-poverty counties only experiences a \$20

absolute income gain relative to the average household in low-poverty counties; this absolute income gain is around 6 percent of the gains that we measure for the expanded CTC (see Appendix Tables B1-B3). It is thus unlikely that the childless EITC expansion meaningfully and differentially affects consumption patterns across high-, medium-, and low-mobility counties. Given that we cannot seasonally adjust our credit card spending data (data for 2019 are not available), differential receipt of the EITC across high-, medium, and low-poverty counties could slightly affect our estimates of consumption on the intensive margin.

Table: Association of county poverty status and gains from the childless EITC expansion

	Absolute \$ Gain due to Childless EITC Expansion	Log \$ Gain due to Childless EITC Expansion	Absolute \$ Gain due to Childless EITC Expansion	Log \$ Gain due to Childless EITC Expansion
Medium Poverty County X Post-Expansion	14.97*** (2.637)	0.001*** (0.000)		
High Poverty County X Post-Expansion	20.28*** (3.576)	0.002*** (0.000)		
Country Poverty Rate X Post-Expansion			184.83*** (22.534)	0.016*** (0.002)

Note: This analysis is limited to counties observed in the CPS-ASEC. “Absolute Gain” refers to the additional dollar value of income due to the childless EITC expansion.

1.2) Many purchases were made online and not in person in 2021. If anything, I'd expect more purchases online for higher income people, which could drive up the relative number of in-person purchases that you'll attribute to the CTC. Is this a threat to identification?

Response: We appreciate the opportunity to clarify this point. We now detail in the manuscript that: For around 60% of establishments, we have data on mean debit/credit card spending for transactions that occur online. Given the imperfect coverage and generally low values of online spending, we do not include these in our primary analysis. In additional analyses, however, we find positive but highly imprecisely estimated differences in online spending amounts for high- and medium-poverty counties relative to low-poverty counties after the CTC expansion relative to before at any of our establishment types. Put differently, we do not find evidence that differential online spending across counties before versus after the CTC reform should bias our findings upwards.

1.3) How do your results compare with "Spending Responses to High-Frequency Shifts in Payment Timing: Evidence from the Earned Income Tax Credit" by Aladangady et al? Be sure to cite this paper along with Goodman-Bacon & McGranahan, 2008.

Response: Aladangady et al. (2018; AEJ forthcoming) explore how shifts in the timing of EITC payments to households affect spending. Using high-frequency spending data coupled with variation in the timing of tax refunds stemming from the 2017 PATH Act, they document considerable consumption responses to a two-week shift in the timing of payments. Their results provide clear evidence that spending out of EITC refunds is not smoothed evenly throughout the year. Evidence from our event study analysis of reduced spending in restaurants and grocery stores in partially treated months also suggests that spending out of CTC refunds is not smoothed evenly throughout the year, even though – compared to Aladangady et al. (2018; AEJ forthcoming) – our results combine the effect of a shift in timing and an increase in the level of the transfer. We acknowledge this link to their paper in the Discussion section.

1.4) Paragraph on page 15 is too long and dense.

Response: We appreciate the reviewer pointing this out and we have now revised the paragraph accordingly.

Comments from Reviewer #2:

The study “Consumption Responses to an Unconditional Child Allowance in the United States” examines consumption responses to the 2021 Child Tax Credit expansion using anonymized mobile-location data and debit/credit card data. This is an innovative study design and the authors well motivate the study’s aims. I have several suggestions that I hope might improve the manuscript.

2.1) The manuscript would be improved if the authors more thoroughly introduced and justified some of their methodological decisions. In particular, the authors should explain why they chose March-May 2022 as “treatment months” for analyzing lump-sum CTC effects on consumption. While issuance timing of the monthly 2021 CTC payments around the 15th of each month from July to December 2021 is clear, there is less certainty about the issuance and receipt timing of lump-sum CTC. Similarly, the authors should also discuss measurement error in their proxy for lump-sum CTC “treatment” (represented by an indicator variable for “lump-sum treatment” months) given that there is considerable variability in the timing and magnitude of lump-sum tax credit receipt, which goes uncaptured by an indicator variable. Several studies present data showing that the timing of lump-sum tax credit receipt is far from uniform around tax time (Jones & Michelmore; Aladgady et al., 2018) and is correlated with household income (Maag, Roll, Oliphant, 2016). The authors should consider how this source of measurement error in their treatment variable could influence their results. (This issue is less concerning for treatment T, a binary indicator of the timing of monthly CTC receipt, since the monthly CTC payments were received around the 15th of each month, and thus, were fairly uniform in terms of both timing and total monthly outlays over the 6-month period).

Response: We appreciate the reviewer's feedback and recommendations for improvement. In the revised manuscript, we now give greater discussion to timing of the lump-sum payments. Specifically, we cite IRS evidence of the distribution of refundable tax credit payments, noting that around half of EITC benefits (a refundable tax credit for which we have available data on payment timing) are paid in the final week of February, while approximately three-fourths are paid out by the end of March, and approximately 90 percent are paid out by the end of April. These estimates are from the Internal Revenue Service: Research, Applied Analytics, & Statistics (as cited in Aladangady et al., 2018). We do not have direct data on the timing or location of the lump-sum CTC distributions in 2022; as such, the reviewer is right that identifying the lump-sum payment timing is far more challenging than isolating the monthly payments. In our primary models, we identify the lump-sum treatment period of March and April 2022. In our revised manuscript, we now also include sensitivity tests in which we identify the lump-sum period as February, March, and April 2022 (results are comparable). Moreover, we have moved up our event study figures to the main text; the event study plots show effect estimates by month and do not force us to directly identify the lump-sum treatment timing. In all cases, we emphasize more to the reader that there is less certainty around issuance and receipt timing for the lump-sum CTC payments.

2.2) The authors do not include county fixed-effects in their regression models, which seem appropriate in order to control for any time-invariant, unmeasured characteristics of the county. (I am especially concerned about unobserved factors that are correlated with poverty status, which the authors use as a proxy for CTC gains.) Can the authors describe why county fixed-effects were omitted from their models? Ideally, the authors could show that their results are robust to the inclusion of county fixed-effects.

Response: In our revised manuscript, we now present findings from several alternative specifications that the reviewers have recommended: (1) our baseline estimates with year/month fixed effects and a new set of controls for county unemployment trends from the Bureau of Labor Statistics, state-month policies on COVID mobility restrictions, state-month availability of SNAP Emergency Allotments, and state-month availability of expanded unemployment benefits; (2) our baseline estimates but also with state fixed effects; (3) our baseline estimates but also with interacted state-year/month fixed effects; (4) our baseline estimates but also with county fixed effects; (5) a model in which we create poverty bins based on the state-specific poverty distribution. Our primary findings regarding increased extensive-margin consumption at childcare centers hold across all five models. That said, our baseline model (the first of the five listed above) more consistently passes our placebo tests and pre-trends analyses and thus remains our preferred specification, as we now clarify in the manuscript. We now show results across each of these models, however, to emphasize the consistency of our primary findings across each. The table below illustrates how we demonstrate this point in the revised manuscript.

Table: Estimated effect of CTC expansion on visits to child care centers by model specification

	Child Care: Pre-Trend (Jan. – June 2020)	Child Care: All Payments	Child Care: Monthly Payments	Child Care: Lump-Sum Payments
Month FE				
Medium Poverty X Treatment	-0.004 (0.002)	0.030*** (0.008)	0.024** (0.008)	0.037** (0.011)
High Poverty X Treatment	0.005 (0.003)	0.029** (0.009)	0.024* (0.011)	0.036** (0.012)
State FE, Month FE				
Medium Poverty X Treatment	-0.003 (0.002)	0.032*** (0.008)	0.029** (0.009)	0.040*** (0.011)
High Poverty X Treatment	0.007* (0.003)	0.032** (0.010)	0.029* (0.012)	0.043*** (0.012)
State x Month FE				
Medium Poverty X Treatment	-0.005* (0.002)	0.027** (0.008)	0.023** (0.008)	0.032** (0.010)
High Poverty X Treatment	0.002 (0.003)	0.031** (0.011)	0.027* (0.012)	0.035** (0.012)
County FE, Month FE				
Medium Poverty X Treatment	-0.002 (0.002)	0.037*** (0.008)	0.036*** (0.009)	0.048*** (0.010)
High Poverty X Treatment	0.010** (0.003)	0.043*** (0.011)	0.040** (0.012)	0.061*** (0.012)

Within-State Poverty Bins (State, Month FE)				
Medium Poverty X Treatment	0.002 (0.003)	0.022** (0.007)	0.023** (0.007)	0.021* (0.009)
High Poverty X Treatment	0.002 (0.002)	0.025** (0.008)	0.027** (0.009)	0.019 (0.010)

Standard errors in parentheses. * $p < 0.05$, ** $p < 0.01$, *** $p < 0.001$

2.3) For identification, the authors “exploit variation in treatment intensity across counties stemming from differences in the income gains after the switch from the regular to the expanded CTC.” The authors use a county’s poverty rate to proxy these CTC gains, and operationalize poverty status as a categorical variable (low, medium, high poverty) in their main models. Results are then presented as effects of the CTC expansion on consumption (visits and spending) among high- and medium- poverty counties relative to low-poverty counties. Estimates for each establishment type are shown in Figures 1 and 2, with overlapping confidence intervals. Point estimates for high- and medium- poverty counties are also quite similar, with sometimes larger point estimates for medium-poverty counties. The manuscript would be strengthened if the authors briefly discussed this pattern of results (given that there appears to be little difference between the high- and moderate- “treatment” intensity groups).

Response: The author is correct that, for many consumption categories, point estimates are similar for medium- and high-poverty counties relative to low-poverty counties. We now discuss in the text that these patterns suggest that the proportional effect of the CTC on consumption may not increase linearly at higher levels of CTC exposure. This further justifies the use of the poverty bins rather than a continuous measure of the poverty rate (though we present these results also in the appendix). For our primary outcome of interest, the findings suggest that increases in visits to childcare centers were comparable for families in medium- and high-poverty counties relative to low-poverty counties.

2.4) The authors include a somewhat limited set of covariates in their model and do not control for other policy changes that occurred over this study period (of which there were many). This manuscript would be strengthened by further justification that these results are not due to other policy changes.

Response: We have now altered our estimation strategy to include state-month variation in policy availability that could otherwise bias our estimates. Specifically, our revised models now include state-month indicators of whether the state provides SNAP Emergency Allotments (which allows states to increase SNAP benefit levels to the maximum benefit amount), and state-month indicators on availability of expanded unemployment benefits (which some states began withdrawing in the summer of 2021). Additionally, we have added in county-month data on

unemployment trends from the Bureau of Labor Statistics, and state-month policies on COVID mobility restrictions from the Oxford COVID-19 Government Response Tracker. These additional covariates help to further account for policy changes that have occurred throughout the study period that may have differentially affected consumption patterns across low-, medium-, and high-poverty counties. Moreover, please see Response 1.1 for a discussion of how we treat the American Rescue Plan's childless EITC expansion.

Comments from Reviewer #3:

In this paper, the authors investigate how changes to the Child Tax Credit (CTC) in tax year 2021 affected expenditure patterns and in-person visits. Their identification strategy relies on comparing cell phone and credit/debit card information across counties categorized as low-, medium-, and high-poverty, assuming higher poverty areas were more treated by the policy change than lower poverty areas. The authors present an extensive set of results based on numerous sample restrictions and specifications. The authors conclude that the expanded CTC resulted in more visits to child care facilities and increased spending at restaurants and grocery stores.

I enjoyed reading this paper and commend the authors on their thorough and careful analysis. This is an important topic and an exciting area of research. I've compiled a list of comments below that I hope will help improve the paper.

3.1. Big picture issues:

a. It is important to spend more time on the mechanisms by which the CTC would lead to higher child care visitations. Can you show that parents were more likely to go back to work in person? It seems like you are implying that the CTC allowed parents to afford child care that they were foregoing previously, the cost of which prevent employment. If not, we are left wondering why more cash on hand would lead to higher child care visits. The discussion section touches on these points but raises more questions than it answers.

Response: Our findings suggest that the CTC contributed to increases in visits to formal child care centers. Households may now afford formal childcare, as either a complement or a substitute to informal childcare. Though an analysis of labor supply decisions is beyond the scope of this study, it is possible that these findings could contribute to explaining the expanded CTC's lack of negative employment effects in 2021 (Ananat et al., 2022; Pilkauskas et al., 2022; Enriquez et al., 2023). First, increases in childcare consumption are not incompatible with the lack of employment effects. Gennetian et al. (2022, 2023) provide evidence that child-related cash transfers lead to increased expenditure in childcare (17%), with no associated employment effects. Second, it is possible that the unconditional cash transfer might have lowered participation or hour constraints for some parents, whilst reducing employment incentives for others. Our findings from the Annual Social and Economic Supplement (ASEC) of the Current

Population Survey (CPS) that, following the CTC expansion, single parents with children living in poorer counties became less likely to report being non-employed or working part-time due to child care problems, suggest that the first channel might be at play, at least for part of the population. Finally, the temporary nature of the expanded CTC might have reduced individual incentives to find a stable job in response to increased cash assistance. Unfortunately, our data do not allow us to uncover the precise mechanisms behind the effects that we document, but these do not seem to be inconsistent with the lack of employment effects. We now discuss these points in the Discussion section.

3.2. A major limitation of the paper is that you do not have expenditure data on child care services. I understand that this is a data issue, but this should be highlighted in both the intro and conclusion.

Response: We acknowledge in the Data and Findings sections that we measure visits to child care centers, but do not have credit/debit card spending data for child care centers. We also more carefully represent our results in the Discussion section. Since we do not summarize our findings in the Introduction, we have not added this information there.

3.3. Another major limitation is that you do not have spending data for 2019, meaning that you are effectively using 2020 as the baseline for any deviations in spending in 2021 and beyond. This seems very problematic.

Response: We acknowledge in the paper that we are not able to seasonally adjust our spending data in the way that we can for our mobility data. We do not use any spending data from 2020 in our analysis (we do not seasonally adjust to a 2020 baseline) given that spending patterns for 2020 are unlikely to be a reliable baseline. The precision of our spending estimates thus relies on an assumption that seasonal variation in spending will apply similarly to medium- and-high poverty counties relative to low-poverty counties. We now acknowledge this in our Data section.

3.4. What is the economic significance of the results? Why should we care if a \$1000 increase in annual income resulted in 6.7 percent more child care centers?

Response: Thank you for raising this point. Rescaling our reduced-form estimates of the effect of the expanded CTC on childcare visits by the size of the income shock allows for simpler benchmarking of our estimates with those of other similar programs. An informative comparison is with estimates of the effect of the Child and Dependent Care Credit (CDCC) on paid childcare participation (Pepin, 2020). The CDCC is a nonrefundable tax credit based on taxpayers' income and childcare expenses, aimed at reducing childcare costs. Using cross-state and cross-family-size variation in the generosity of the CDCC induced by a 2003 CDCC expansion, Pepin documents that a 10 percent increase in CDCC benefits increases annual paid child care participation by 4–5 percent (elasticity of 0.5) among households with children younger than 13 years old. Our results imply an elasticity of 0.9, indicating larger responses to the expanded CTC. Two factors are likely to explain the larger elasticity in our context: (i) in-person visits capture both participation and

intensive-margin responses; (ii) responses to the CTC may be larger due to its refundable nature, which implies that low-income families are also largely treated. We now clarify this point in the Findings section.

3.5. A better job should be done to synthesize the different, and sometimes contradictory, results. As it currently stands, the authors present many results, most of which are insignificant or inconclusive, and do their best to weave a narrative between them. It leaves the reader wondering how much confidence should be placed in the findings.

Response: We have revised the text to describe our results in a synthetic, yet precise manner in the manuscript, giving more prominence to our primary result of in-person visits to child care centers.

3.6. Identification strategy and methodology: The baseline specification must include state fixed effects. In fact, some might say that state-by-month fixed effects would be more appropriate, though I think that such a specification would be suitable for a robustness check.

Response: In our revised manuscript, we now (1) test the sensitivity of our results across a range of alternative model specifications, including those that the reviewer suggests and (2) give greater clarity to why we prefer our baseline estimates with year/month fixed effects (plus a new set of controls for county unemployment trends from the Bureau of Labor Statistics, state-month policies on COVID mobility restrictions, state-month availability of SNAP Emergency Allotments, and state-month availability of expanded unemployment benefits).

Regarding the first point, Appendix G (see also the Table provided in response to Reviewer 2) now validates the consistency of our primary findings across several different model specifications: our baseline model, a model with state FE, a model with state by year-month FE, a model with county FE, and a model that creates poverty bins based on the state-specific poverty distribution.

Regarding the second point, we elaborate on why we prefer our baseline model that excludes state fixed effects. The primary challenge of including state fixed effects is that states vary widely in their balance of low-, medium-, and high-poverty counties. When creating poverty bins based on the poverty rate distribution of all counties, Mississippi, for example, has only 3 low-poverty counties but 67 high-poverty counties. Rhode Island, conversely, has no high-poverty counties. Simply including state fixed effects effectively forces our model to compare most of Mississippi's counties to a reference group of merely three counties, and offers no full comparison in states such as Rhode Island. A stronger approach to avoid losing information from each state is to create state-specific poverty terciles across each state's counties, as presented in the fifth row of Tables G3 to G5. This approach ensures that one-third of Mississippi's counties are identified as each of low-, medium-, high-poverty counties, avoiding the small-reference-group challenge documented above. The conceptual challenge that remains is that, using this approach, relatively *low*-poverty counties in Mississippi (to continue the example) have a mean poverty rate of 17.9%, which is

higher than the mean poverty rate of 12% in relatively *high*-poverty counties in Connecticut. If the purpose of our model is to apply a treatment that aligns well with gains from the 2021 CTC expansion, then this within-state approach objectively performs worse than our baseline model. Given that the more restrictive model with state fixed effects (or state by year-month fixed effects) more strongly excludes omitted variable bias, however, we also corroborate our primary findings with results from these specifications in Appendix G. Our primary findings from the baseline models of increased childcare consumption holds across all model specifications evaluated.

3.7. State-level policies that affected local economic activity were in flux during the period of analysis. Lockdowns, school policy, mask mandates, unemployment insurance, and so on. All standard errors should be clustered at the state level.

Response: All of our estimates (in the original and revised manuscripts) cluster standard errors at the state level, which we now acknowledge in the revised text. Moreover, we now also include (1) state-month indicators of stringency of COVID containment and health measures from the Oxford COVID-19 Government Response Tracker, and (2) state-month indicators of availability of SNAP Emergency Allotments and expanded unemployment benefits.

3.8. I had to read the description of the main specification multiple times. Specifically, saying that you have an indicator of the poverty rate is confusing. It would be better to say an indicator of the poverty rate tercile, or something similar.

Response: We agree with the reviewer and we now clarify that we segment counties into poverty terciles or poverty bins.

3.9. The most convincing results, by far, were the even study plots and should be presented as the main results. Not only do these charts help support the parallel trends assumption, but they also avoid the awkward (and somewhat arbitrary) monthly groupings.

Response: We agree with the reviewer's suggestion to bring the event study plots into the main analysis. We now give greater discussion to these figures in the Findings section.

3.10. Specifically, the option to take monthly checks or the lump sum payment makes identification trickier than it would be otherwise. Plus, refunds could have been received before March 2022. Avoid all of this by simply using the event study specification which doesn't assume that certain months are more/less treated. Let the data show you dynamics.

Response: We agree with the reviewer's suggestion to bring the event study plots into the main analysis. We also include alternative operationalizations of our lump-sum treatment effects (see response to reviewer comments 2.1 and 2.2).

3.11. If you want a nice simple DD estimate, perhaps you could end the panel in Dec 2021 and just use the average estimate for the second half of 2021.

Response: We now clarify in the manuscript that our estimates for “Monthly payments” (as visualized in Figure 1, for example) provide this exact estimate.

3.12. Perhaps I missed it: why does Figure F1 not start before January 2021? The spending data is limited to 2021 and beyond but not the cell phone data, correct?

Response: Per the reviewer’s request, we have included a version of the event study plot for our mobility estimates that begins in July 2020 (one year before the start of the monthly CTC payments). We prioritize our analyses that use January to June 2021 as the pre-treatment reference period, as the second half of 2020 was more fraught with COVID cases and deaths, continued school closures, the expiration of the Pandemic Unemployment Compensation, and more. Thus, in our primary estimates and the event study we include in the main part of the manuscript, we continue with the presentation of results that starts in January 2021. In the Appendix, we have included the figure copied below, which begins the time series at July 2020 instead. The figure demonstrates that our primary finding regarding increased visits to child care centers after the CTC payments still holds when extending the pre-treatment period.

Figure: Extended event study estimates of the effect of expanded CTC on in-person visits to different establishment types in high-poverty counties relative to low-poverty counties (pre-treatment period beginning in July 2020)

3.13. A more recent poverty rate should be used as a robustness check. Why not use 2018 ACS data to define the poverty terciles? 2010 data from Op. Insights seems like an odd choice.

Response: We agree with the reviewer’s recommendation and appreciate the suggestion. We have now replaced our poverty indicators with 5-year averages of ACS county poverty rates spanning 2014-2018. We avoid any data from 2019-2021 given biased response rates and potentially-altered poverty rates due to the COVID-19 pandemic. The figure below, provided for the reviewers, demonstrates that our 2010 and 2018 poverty rates are strongly correlated, which is why our results do not meaningfully differ after updating the poverty indicators.

Figure: Comparison of county poverty rates: 2014-2018 estimates (new measure used in the paper) versus 2010 estimates (measure used in prior draft)

3.14. More should be said about the variation in COVID policies that could be contemporaneous with the monthly CTC checks. If low-poverty areas, which are concentrated in the south (Figure G1), were treated to different COVID policies in the second half of 2021 this would bias the estimates. State-by-month fixed effects would be the most straightforward way to address this concern and would avoid time-intensive data collection.

Response: The reviewer raises an important point regarding state-month variation in COVID restrictions, which we now address in two ways. First, in all of our estimates, we now include data from the Oxford COVID-19 Government Response Tracker on state-month COVID restrictions. Specifically, we include the indicators of *stringency* (the strictness of ‘lockdown style’ policies that primarily restrict people’s behavior) and *containment and health measures* (capturing ‘lockdown’ restrictions and closures with measures such as testing policy and contact tracing, short term investment in healthcare, as well investments in vaccines). Second, we also include estimates that include state-by-year/month fixed effects, as discussed above. Our primary findings hold when including these interacted state-month fixed effects.

3.15. The first paragraph of the introduction feels very out of place and disrupts the flow of the entire paper.

Response: We have now moved this paragraph down per the reviewer’s request.

3.16. My prior is that lower-poverty areas have less remote-friendly work options. Can this be discussed as it relates to child care? This could be relevant for return-to-work considerations that occurred around the time that the CTC was expanded.

Response: We agree with the reviewer’s suggestion to discuss remote work in relation to our child care findings. Evidence from the Current Population Survey demonstrates that employed adults in higher-poverty counties are less likely to have the option to work remotely relative to employed adults in lower-poverty counties. The binned scatterplot below demonstrates this point, though only among the small subset of counties (n=280) identifiable in the monthly CPS files. Specifically, the scatterplot shows a negative correlation ($r = -0.24$) of the county’s poverty rate (5-year average of 2014-2018) and the share of employed adults with the ability to work remotely (average over January to June 2021). We mention this relationship in our Discussion section. Specifically, we note the likelihood that workers in higher poverty counties are less able to have the option to work remotely and, thus, more likely to need child care support in order to continue working. It is possible that this fact contributes to the larger increase in child care consumption for higher poverty counties after the introduction of the CTC, though we cannot directly evaluate this given the small sample of counties available in the Current Population Survey data (the dataset in which we can estimate remote work shares).

Figure: County poverty rate by remote work possibility among employed adults

Note: County poverty rates are 5-year averages from the American Community Survey. The share of employed adults able to work remotely is from the Current Population Survey (CPS) Basic Monthly files (January to June 2021). The sample is limited to the 280 identifiable counties in the CPS files.

3.17. It would be wise to downplay or cut the “cover more than 328 million U.S. residents (99.6 percent of the population)” comment. It oversells the data you actually use for the analysis, which is likely not representative.

Response: We have now removed this wording to avoid overselling the coverage of our data.

3.18. I liked many of the robustness checks, especially the school falsification check. Can you do the same with public parks? Other places that families might frequent?

Response: We have explored the possibility of adding falsification tests beyond schools. The closest option to the reviewer’s recommendation is NAICS code 712190, or “Nature Parks and Other Similar Institutions.” However, this category is not restricted to free, public parks; instead, it also includes attractions with entrance fees, such as bird or wildlife sanctuaries, natural wonder tourist attractions (e.g., caverns, waterfalls), conservation areas, nature centers or preserves, and national parks. Therefore, it does not meet the same standards as public schools for use in a falsification test.

3.19. Even though I prefer the simple figures, the full regression tables should be included in the appendix.

Response: We now include the regression tables in Appendix G.

3.20. More attention should be given to the potential labor supply effects, and therefore family income consequences, of an unconditional, refundable CTC (see Han, Meyer, and Sullivan, 2022).

Response: We now include a discussion of the existing evidence on labor supply effects in the Discussion section.

3.21. I would recommend reorganizing the paper to demonstrate the validity of the parallel trends assumption before you get into the main results. Again, this can all be accomplished by focusing on the event study results.

Response: We now illustrate our event-study results in our primary Findings section and highlight that they provide reassuring evidence of the validity of the parallel trends assumption. We opted for leaving the discussion of measurement validation, identification tests and robustness checks after the illustration of the main findings, since we felt this would make the manuscript easier to follow.

3.22. An appendix section detailing the visitation data should be included. Most readers will not be intimately familiar with these data.

Response: We have now added an appendix that provides further details on the SafeGraph data (see Appendix C).

3.23. I am not sure I fully agree that the results are understated due to differential cell phone use. Couldn't families with cell phones also have a differential response to the CTC?

Response: Thank you for raising this point. Our statement assumes that families with and without smartphones have similar preferences for consumption (of childcare and other items). If families without smartphones have different consumption preferences (e.g. are averse to the use of childcare), then we would be overestimating the effect of interest. We better qualified our statement in the manuscript.

3.24. Unemployment data at the county level is available from the BLS and should be included as controls. Or at least justify the use of Op. Insights data.

Response: We have followed the reviewer's suggestion to use Bureau of Labor Statistics county-level unemployment data in all estimates, rather than the Opportunity Insights data used before. This gives us full county-month coverage, and we now include the unemployment data as a control in our baseline models. We again thank the reviewer for the useful set of recommendations.

Decision Letter, second revision:

20th July 2023

Dear Dr. Parolin,

Thank you once again for your revised manuscript, entitled "Consumption Responses to an Unconditional Child Allowance in the United States," and for your patience during the re-review process. I am sorry for the delay in sending this decision.

Your manuscript has now been evaluated by the same reviewers who evaluated your original manuscript. All reviewer feedback is included at the end of this letter. Although the reviewers found your manuscript to have improved during revision, Reviewer 3 also raises some important outstanding concerns. We remain very interested in the possibility of publishing your study in Nature Human Behaviour, but would like to consider your response to these outstanding concerns in the form of a revised manuscript before we make a decision on publication.

We ask that you carefully address each of Reviewer 3's remaining points, including:

- 1) Addressing reviewer concerns about the optimal choice of baseline model specifications

- 2) Carrying out the additional robustness checks recommended by Reviewer 3 and including the additional supplementary results requested
- 3) Addressing reviewer concerns about the clarity of the mechanisms, both through additional analyses and discussion
- 4) Ensuring that methodological details and conceptual arguments are clear.

Please note that Reviewer 3 has raised concerns regarding the structure of your manuscript. We are bound by a strict format, which requires that sections appear in the order Introduction, Results, Discussion, Methods. However, we ask that you revise within the confines of this format to ensure that your manuscript is as clear as possible. The end of the Introduction and/or start of the Results section should contain enough of an overview of the data and methodology that readers will be able to understand the results and discussion sections. Further details should be left for the Methods section.

Finally, your revised manuscript must comply fully with our editorial policies and formatting requirements. Failure to do so will result in your manuscript being returned to you, which will delay its consideration. To assist you in this process, I have attached a checklist that lists all of our requirements. Please note in particular that tables in the main text will need to include exact p-values and confidence intervals. (Asterisks to denote statistical significance should not be used as a substitute for actual p-values.) If you have any questions about any of our policies or formatting, please don't hesitate to contact me.

In sum, we invite you to revise your manuscript taking into account all reviewer and editor comments. We are committed to providing a fair and constructive peer-review process. Do not hesitate to contact us if there are specific requests from the reviewers that you believe are technically impossible or unlikely to yield a meaningful outcome.

We hope to receive your revised manuscript within 8 weeks. I would be grateful if you could contact us as soon as possible if you foresee difficulties with meeting this target resubmission date.

- Include a "Response to the editors and reviewers" document detailing, point-by-point, how you addressed each editor and referee comment. If no action was taken to address a point, you must provide a compelling argument. This response will be used by the editors and reviewers to evaluate your revision.
- Highlight all changes made to your manuscript or provide us with a version that tracks changes.

[REDACTED]

We look forward to seeing the revised manuscript and thank you for the opportunity to review your work. Please do not hesitate to contact me if you have any questions or would like to discuss these revisions further.

Sincerely,
Aisha

Aisha Bradshaw, PhD
Senior Editor
Nature Human Behaviour

Reviewer expertise:

Reviewer #1: economics, tax policy

Reviewer #2: public policy, tax credits

Reviewer #3: family economics

REVIEWER COMMENTS:

Reviewer #1:

None - This reviewer provided comments only the editors, indicating that they are satisfied with your revision.

Reviewer #2:

Remarks to the Author:

I appreciate the authors' careful responses to my initial comments. The revisions have significantly improved the paper. This study makes a valuable contribution to the literature. Thank you for considering my feedback.

Reviewer #3:

Remarks to the Author:

I enjoyed reading the revised manuscript and commend the authors on a thorough response. Aside from the unusual organization (more on this below), the paper was improved in almost every dimension. I will mostly restrict my comments to those related to my first round of comments. I have a few, new comments at the end, however, that couldn't be overlooked.

1. Big picture issues

a. Mechanisms: I agree that an absence of detectable labor supply responses is not inconsistent with increased formal child care spending, yet even with the cited papers this issue is far from settled and there are other papers suggesting non-trivial labor supply responses to the policy (for example, see Corinth, Meyer, Stadnicki, and Wu, 2021). I understand that this is beyond the scope of the paper, yet it does leave the reader wondering what is actually going on.

i. Can you show the number of child care establishments over time across the poverty bins? There were many reports of child care centers closing during COVID, and if closures and re-openings are correlated with poverty then this could be part of the story. This may help you shed light on this question: did parents substitute across care types, switching from family providers to centers that show up in your data?

ii. Another suggestion that would give readers a better sense of the combined response to the CTC is to make the following back-of-the-envelope calculation: if the CTC increased restaurant spending by 1.9 percent and increased grocery spending by 2 percent (line 142) then how much could child care expenditures have increased? For example, if high-poverty residents received \$500 more per month, could you say how much of that \$500 went towards increased restaurant spending and groceries, thereby allowing you to estimate a rough amount that could have been spent on child care?

b. I went to the SafeGraph Spend website and it does appear that there are observations in the expenditure data for child care centers. Perhaps this is not true in your data? Either way, I suspect these observations are very limited, yet you should still discuss if the patterns hold when using child care expenditures, if possible. In other words, create a Figure E1 for whatever child care expenditure data you possess.

i. The spending and visitation connection is arguably weaker for child care than other service industries. I go to a store to spend money, so that correlation is strong, whereas the connection between payment and visits in child care is buffered by different contracts and the interaction with school and work schedules.

2. Identification strategy and methodology

a. I appreciate the robustness checks using various kinds of robustness checks. The stability of the estimates will give readers more confidence in the main results. Although I understand your argument for how different fixed effects will shift the focus of the identifying variation, I still strongly disagree with the choice to omit any location fixed effects from baseline specification. This is not standard and raises all sorts of questions. The fact that your pre-trends look better in the specification that omits the fixed effects is not a valid justification for using it over the more standard specification; if anything, that may actually raise more concerns. The good thing is that your estimates are stable. You should make the state and month fixed effects specification your baseline, after which you can demonstrate how county fixed effects, state-by-month fixed effects, and within-state rankings don't drastically change your findings.

i. On this point, you should probably show the event studies for all of these robustness checks in the appendix, not just the average point estimates.

b. The description of the main specification is improved, yet the vector (X) of controls is still confusing. Why control for log population density but levels of COVID cases? Are unemployment "trends" (line 384) trends or the unemployment rate? There is almost no explanation of how the Oxford data is utilized. We need much more information on this. As of now we must take it on faith that you are adequately controlling for COVID-related factors.

i. You could lean into the state-by-month fixed effects to say that, even if you did not adequately control for COVID-related factors with the Oxford data, those factors should be captured by the state-by-month fixed effects.

ii. Why not include the coefficients on some of these additional controls, at least in the appendix? They will help the reader get a sense for if they were independently affecting the outcome variables. As of now, we have very little idea how things are related.

3. Other assorted comments

a. I understand the response about some parks having a fee, but you explicitly dropped public places (Line 256). Can't you use places that are unlikely to have a fee as another placebo check?

b. Line 433. This doesn't seem like a reasonable assumption. If anything, it seems like consumption preferences would differ between cell phone and no-cell phone families. If this is the case, you may actually be overstating the results, not understating them. Absent some supporting evidence, it might be wise to just cut the sentence.

4. A few new comments

a. The organization of the paper is confusing. I did see that the "results-before-methodology" change is due to journal policies, but there were other changes that also make the manuscript harder to read. For example, the discussion should come after we are familiar with the methodology and data. It would be much better to mirror the first manuscript's organization as close as possible while still checking whatever boxes are necessary at the journal.

b. It would be more convincing if you control directly for county-level wages, even if they are imperfect (see the QCEW). Combined with the unemployment rate, this would go a long way to convincing readers that you are not attributing other economic factors to the CTC response.

c. Line 192: I'm not sure most people considered the policy change temporary (many assumed it would be made permanent). But let's say they did assume it was temporary: why would it reduce incentives to find stable work? Are you referring to people on the margin who couldn't afford formal care without the more generous credit? This should be clarified or the sentence could be deleted.

d. Please include the event study of the school visit placebo in the appendix. This is an important outcome that will be on the minds of a lot of people so you should emphasize it more. In fact, it would be wise to show how your results change when you omit vs include this as a control variable (line 386).

e. The pre-trend checks are fine, even if they usually turn out to be null due to reduced sample sizes. Most of the action we are concerned about comes later in the year as schools and local economies adjusted to the later stages of COVID. What I'm getting at is distinguishing between pre-trends biasing your estimates and contemporaneous factors correlated with poverty bins.

f. Upon closer inspection, I am confused as to the grouping of cars, clothes, hobbies, and home expenditures. These should be split up into more than one category. There are more than enough observations to do so. Cars and home improvement supplies were greatly affected by supply chains while clothing and hobbies were not. There may be different propensities to spend the CTC on those subcategories, too. It isn't that surprising to find a null effect given this random grouping. If there is a better justification for grouping those categories together, please include it.

Author Rebuttal, second revision:

Response to Reviewers #2, "Consumption Responses to an Unconditional Child Allowance in the United States" at Nature Human Behaviour

We greatly appreciate the useful set of suggestions from the editors and Reviewer 3. We also thank Reviewers 1 and 2 for the positive feedback and their support of our manuscript. In this response letter,

we first outline the broad changes made to our revised manuscript while responding to the requests of the editorial team. We then proceed to a point-by-point reply to Reviewer 3's comments.

Comments from Editorial Team:

1) Address reviewer concerns about the optimal choice of baseline model specifications.

Response: We have now adopted the reviewer's recommendation and prioritize the models with state and month fixed effects. We have also added in the county-level wage controls that the reviewer requested. As before, we continue to show in sensitivity tests results from each successive specification in the Appendix. Our primary findings regarding increased child care center and food consumption are broadly consistent across specifications. We again thank the reviewer for clear guidance and suggestions on the decision of which specification to prioritize.

2) Carry out the additional robustness checks recommended by Reviewer 3 and include the additional supplementary results requested.

Response: We have now carried out the additional robustness checks (including requested event studies and alternative specifications) that the reviewer has encouraged us to include. This includes an additional placebo test for visits to religious institutions; similar to our 'schools' placebo test, we find no evidence of CTC-induced increases in visits to places of worship. This test provides further credibility of the mobility data that underpin our core findings.

3) Address reviewer concerns about the clarity of the mechanisms, both through additional analyses and discussion.

Response: We have added several analyses per the reviewer's recommendation, including analyses of child care openings by time and poverty group, the association of child care credit/debit card spending, and more. We discuss these below in our point-by-point responses.

4) Ensure that methodological details and conceptual arguments are clear.

Response: We have now addressed each of the points on methods and concepts that the reviewer marked as unclear. This includes elaboration on our control variables, clarifying our identification strategy before presenting results, clarifying when we use (un)logged indicators and showing that these decisions do not meaningfully affect our results, and more. We discuss these below in our point-by-point responses.

Please note that Reviewer 3 has raised concerns regarding the structure of your manuscript. We are bound by a strict format, which requires that sections appear in the order Introduction, Results, Discussion, Methods. However, we ask that you revise within the confines of this format to ensure that your manuscript is as clear as possible. The end of the Introduction and/or start of the Results section should contain enough of an overview of the data and methodology that readers will be able to understand the results and discussion sections. Further details should be left for the Methods section.

Response: Thank you for the guidance in approaching this restructuring. We have now added necessary details for understanding our findings to the start of the Results section. We add greater detail on our data and identification strategy there before presenting our findings.

Finally, your revised manuscript must comply fully with our editorial policies and formatting requirements. Failure to do so will result in your manuscript being returned to you, which will delay its consideration. To assist you in this process, I have attached a checklist that lists all of our requirements. Please note in particular that tables in the main text will need to include exact p-values and confidence intervals. (Asterisks to denote statistical significance should not be used as a substitute for actual p-values.) If you have any questions about any of our policies or formatting, please don't hesitate to contact me.

Response: In our revision, we made sure to comply fully with the journal's editorial policies and formatting requirements. In particular, we devoted special attention to including exact p-values and confidence intervals.

Comments from Reviewer #3:

I enjoyed reading the revised manuscript and commend the authors on a thorough response. Aside from the unusual organization (more on this below), the paper was improved in almost every dimension. I will mostly restrict my comments to those related to my first round of comments. I have a few, new comments at the end, however, that couldn't be overlooked.

Big picture issues

3.1. Mechanisms: I agree that an absence of detectable labor supply responses is not inconsistent with increased formal child care spending, yet even with the cited papers this issue is far from settled and there are other papers suggesting non-trivial labor supply responses to the policy (for example, see Corinth, Meyer, Stadnicki, and Wu, 2021). I understand that this is beyond the scope of the paper, yet it does leave the reader wondering what is actually going on.

Response: We thank the reviewer for pointing out the paper by Corinth et al. (2021), which we now reference in the paper. Our reading of the existing literature on the employment effects of the CTC is that analyses relying on reduced-form estimates of the CTC effects shows no consistent negative employment responses (see Enriquez, Jones and Tedeschi, 2023, for a review). Papers such as Han, Meyer and Sullivan (2022) and Corinth et al. (2021) rely instead on empirical simulations of potential labor supply responses based on participation elasticities from prior work. As we highlight in the paper, key for the interpretation of our findings is the observation that the childcare effects that we document are not incompatible with the lack of a detectable employment

response. Even absent employment effects, an increase in income may affect the quantity and mode of childcare demanded by households with children.

3.2. Can you show the number of child care establishments over time across the poverty bins? There were many reports of child care centers closing during COVID, and if closures and re-openings are correlated with poverty then this could be part of the story. This may help you shed light on this question: did parents substitute across care types, switching from family providers to centers that show up in your data?

Response: Thanks for this suggestion. Below, we provide a figure that tracks the share of formal child care centers that are likely closed (at least a 75% decline in in-person visits in a month relative to the same month in 2019) by the poverty status of the county over time. Individual establishments can stay in the SafeGraph data even if they close for certain months, so tracking the closure rates is more reliable than simply tracking the number of establishments in the dataset. The data suggest no meaningful differences in closure rates by poverty level throughout 2021 compared to July-December 2021 (post-CTC). A statistical analysis confirms this, as we show in the table below. Based on this evidence, it is unlikely that our main findings are driven by differential re-opening rates across poverty bins.

Figure: Trends in share of formal child care centers likely closed by county’s poverty status

Table: Estimate of share of formal child care centers likely closed by county’s poverty status and timing

	Share of Child Care Centers Likely Closed
Medium Poverty Counties	0.005 (0.004)
High Poverty Counties	-0.001 (0.006)
Medium Poverty Counties X CTC Months	0.000 (0.004)
High Poverty Counties X CTC Months	0.007 (0.005)
N	2,175,357

Standard errors in parentheses; * $p < 0.05$, ** $p < 0.01$, *** $p < 0.001$

3.3. Another suggestion that would give readers a better sense of the combined response to the CTC is to make the following back-of-the-envelope calculation: if the CTC increased restaurant spending by 1.9 percent and increased grocery spending by 2 percent (line 142) then how much could child care expenditures have increased? For example, if high-poverty residents received \$500 more per month, could you say how much of that \$500 went towards increased restaurant spending and groceries, thereby

allowing you to estimate a rough amount that could have been spent on child care?

Response: Unfortunately, our data are not structured in a way that make it possible to reliably provide such an estimate. Our spending is place based, not individual based, so we cannot produce estimates of total spending per person at baseline that could be used to contextualize the findings in this way. At most, we could produce an estimate of “mean monthly spending per customer at specific establishments” at the county level, but these values are likely to understate levels of individual spending that are needed to meet the reviewer’s request. Consider the following example. Imagine a county with 2 stores: WalMart and FoodMart. If a person shops twice for groceries at WalMart, spending mean of \$20, and twice at FoodMart, spending mean of \$24, we capture that mean spending at grocery stores was \$22 per transaction and \$48 total per customer, though in reality this customer spent \$88 total at grocery stores that month, roughly double the mean value that our placed-based data would display. Our mean monthly spending per customer is biased downward if this person splits total monthly transactions at different establishments. Thus, any attempt to estimate individual-level monthly spending using mean spending per customer will likely be understated, as one person counts as two customers in our dataset in this example: a customer at WalMart and a customer at FoodMart.

3.4. I went to the SafeGraph Spend website and it does appear that there are observations in the expenditure data for child care centers. Perhaps this is not true in your data? Either way, I suspect these observations are very limited, yet you should still discuss if the patterns hold when using child care expenditures, if possible. In other words, create a Figure E1 for whatever child care expenditure data you possess.

Response: Child care centers do exist in the expenditure data, but they are too small in quantity to be used reliably in analyses such as ours. Consider that there are only 82 counties with child care centers present in the debit/credit card data in a given month (compared to 2,774 counties with restaurants present in the expenditures data, for example); most of these counties only include one child care center. The card-based estimates for child care centers are too noisy to provide useful information. See the example below from an event study plot of our main estimates, for example (in which the Y-axis is 10 times the width of the Y-axis used in our primary analyses).

Figure: Event study specification of child care spending (82 counties) with enlarged Y-axes

We have also followed the reviewer’s suggestion to replicate Figure E1 for child care spending. Recall that Figure E1 is titled “Is mobility data a useful proxy for consumption on the extensive margin? Binned scatterplots of county-level, monthly in-person visits to establishments (X-axis) and transactions with debit/credit cards (Y-axis).” Replicating this analysis for child care spending does show a positive, near-linear association between monthly number of transactions at child care centers (among counties where such information is available) and monthly visits to child care centers. This provides further confidence that our measure of consumption on the extensive margin (visits) is a strong proxy of actual consumption.

Figure: Replication of Figure E1 for child care visits and transactions (82 counties); binned scatterplots of county-level, monthly in-person visits to establishments (X-axis) and transactions with debit/credit cards (Y-axis).

3.5. The spending and visitation connection is arguably weaker for child care than other service industries. I go to a store to spend money, so that correlation is strong, whereas the connection between payment and visits in child care is buffered by different contracts and the interaction with school and work schedules.

Response: We share the reviewer's view. People usually do not swipe their credit card every time they drop off their children, and potentially pay for the care services with means other than credit cards. This helps to explain why we have very high data coverage for visits to child care centers, but very low data coverage for credit/debit card spending at child care centers (see discussion above). We believe this justifies using mobility data rather than spending data to estimate child care usage, since the former are likely a more accurate indicator of consumption (even though, as we show above, the two are correlated). At the same time, the increase in visits to care centers would likely not be possible without an increase in spending. The consistency of the timing of increases in visits to childcare centers with the months when CTC payments were delivered suggests that the increased spending occurred in those months, leading to greater use of center-based care.

Identification strategy and methodology

3.6. I appreciate the robustness checks using various kinds of robustness checks. The stability of the estimates will give readers more confidence in the main results. Although I understand your argument for

how different fixed effects will shift the focus of the identifying variation, I still strongly disagree with the choice to omit any location fixed effects from baseline specification. This is not standard and raises all sorts of questions. The fact that your pre-trends look better in the specification that omits the fixed effects is not a valid justification for using it over the more standard specification; if anything, that may actually raise more concerns. The good thing is that your estimates are stable. You should make the state and month fixed effects specification your baseline, after which you can demonstrate how county fixed effects, state-by-month fixed effects, and within-state rankings don't drastically change your findings.

Response: We have adopted the reviewer's recommendation and now prioritize the models with state and month fixed effects. We have also added in the county-level wage controls that the reviewer requests below. As before, we continue to show in sensitivity tests results from each successive specification in the Appendix. Our primary findings regarding increased child care center and food consumption are generally robust to different specification choices. We again thank the reviewer for clear guidance and suggestions on the decision of which specification to prioritize.

3.7. On this point, you should probably show the event studies for all of these robustness checks in the Appendix, not just the average point estimates.

Response: We have added the event studies for our focal categories in each robustness check to our Appendix. This is in addition to the table-based presentation of the regression results.

3.8. The description of the main specification is improved, yet the vector (X) of controls is still confusing. Why control for log population density but levels of COVID cases? Are unemployment "trends" (line 384) trends or the unemployment rate? There is almost no explanation of how the Oxford data is utilized. We need much more information on this. As of now we must take it on faith that you are adequately controlling for COVID-related factors.

Response: We agree with the reviewer and have now added more information on each indicator. Regarding the Oxford data, we have now added into the manuscript a longer description that we previously removed when dropping all footnotes due to format requirements. Specifically, we detail that we include the Containment and Health Index measure from the Oxford COVID-19 Government Response Tracker. This summarizes data on 13 indicators in total (including nine from its 'Stringency Index') capturing state-level policies related to: workplace closures, cancellation of public events, restrictions on public gatherings, closures of public transport, stay-at-home requirements, public information campaigns, restrictions on internal movements, international travel controls, testing policy, extent of contact tracing, face coverings, and vaccine policy. The index takes the mean score across the normalized distributions of each metric to produce a single statistics taking value between 0 and 100, with a higher score implying stricter containment measures.

Regarding the use of levels or logs of COVID cases: there are some county-months with no reported COVID cases (concentrated in counties with small population sizes; 0.5% of all

county-months), so we use level of cases per capita in our primary analysis to maintain these county-months. Below, we show a sensitivity test in which we apply log COVID cases per capita and exclude such counties; the results are consistent. In the table below, we also test whether our estimates are sensitive to applying *levels* of county unemployment rates and log mean wages, or *within-county variation* (or, *demeaned levels*) in unemployment and mean wages. We prioritize levels in our base analysis, but results are consistent either way.

As discussed in our reply to the reviewer’s next comment, we also present models with state-year fixed effects and county fixed effects, which also effectively account for place- and time-based variation in COVID-related factors.

Table: Estimates of CTC’s effects on visits to child care centers by model specification

	Levels of COVID Cases + Levels of County Unemployment/Wages	Log of COVID Cases + Levels of County Unemployment/Wages	Levels of COVID Cases + Within-County Variation in Unemployment/Wages
Medium Poverty Counties	-0.004 (0.012)	-0.006 (0.012)	-0.002 (0.011)
High Poverty Counties	0.007 (0.015)	0.006 (0.015)	0.010 (0.013)
Medium Poverty Counties X CTC Months	0.032*** (0.008)	0.034*** (0.008)	0.033*** (0.008)
High Poverty Counties X CTC Months	0.033** (0.010)	0.036*** (0.010)	0.032** (0.010)
Medium Poverty Counties X Partially Treated Months	0.037*** (0.010)	0.040*** (0.010)	0.038*** (0.010)
High Poverty Counties X Partially Treated Months	0.019 (0.013)	0.023 (0.013)	0.021 (0.012)
County Unemployment Rate (level)	-0.008* (0.003)	-0.007* (0.003)	
County Mean of Weekly Wage during Quarter (log)	-0.127*** (0.031)	-0.126*** (0.031)	
County Unemployment Rate (demeaned)			0.003 (0.005)
County Mean of Weekly Wage during Quarter (demeaned)			-0.000 (0.000)
COVID Cases per Capita (levels)	0.007 (0.005)		0.009 (0.005)
COVID Cases (log)		0.024	

(0.018)

N	47464	47226	47464
----------	-------	-------	-------

All models include state and year-month fixed effects, and controls for log population density, year-over-year changes in in-person school visits, state-months without SNAP Emergency Allotments, state-months without Unemployment Insurance supplements, and Oxford Government Tracker's COVID health containment measure index. Standard errors in parentheses; * $p < 0.05$, ** $p < 0.01$, *** $p < 0.001$

3.9. You could lean into the state-by-month fixed effects to say that, even if you did not adequately control for COVID-related factors with the Oxford data, those factors should be captured by the state-by-month fixed effects.

Response: We agree and now emphasize this point after documenting the state-level controls above and when presenting our state-by-month sensitivity tests.

3.10. Why not include the coefficients on some of these additional controls, at least in the Appendix? They will help the reader get a sense for if they were independently affecting the outcome variables. As of now, we have very little idea how things are related.

Response: We have added a table showing all coefficients in the Appendix.

Other assorted comments

3.11. I understand the response about some parks having a fee, but you explicitly dropped public places (Line 256). Can't you use places that are unlikely to have a fee as another placebo check?

Response: We have now added another placebo test using visits to religious institutions, and we have clarified the line about removing public places in the manuscript. We discuss these in turn.

First, we revisited all available NAICS codes to find a plausible category of establishments to use as an alternative placebo test (a group of places that should not be influenced by increased cash available due to the CTC payments). Religious organizations (NAICS code 813110, which includes churches, synagogues, mosques, temples, and other places of worship) stood out as a credible candidate. Similar to schools, we should not expect that CTC-driven increases in income meaningfully affect seasonally-adjusted visits to places of worship. We show the placebo test below, and have also added this into the manuscript. Across each payment type and both treatment types (poverty bins and continuous poverty rate), we find no significant effect on visits to places of worship, with point estimates generally hovering around zero. Alongside our placebo test for schools, we view this as strong evidence that our mobility data is not systematically picking up alternative factors affecting movement that would bias our results.

Figure: Placebo test: Effects of the CTC payments on seasonally-adjusted visits to religious institutions

We have also clarified the line about removing public places. There are some NAICS categories, such as “712190 - Nature Parks and Other Similar Institutions,” that feature a mix of for-cost and no-cost establishments with no straightforward way of disaggregating the two (we do not have a more-precise NAICS code to distinguish among this group). We have altered the line referenced (previously line 256) to clarify this point: we do not include ‘Nature Parks’ in our Family Entertainment category because it includes many public and no-cost places, yet the category still includes too many for-cost places to use as a placebo test comparable to our ‘schools’ test. Consider some of the examples provided in the public list of places in this NAICS code: for-cost zoos, See Rock City, Attractions Hawaii, Mazza Vineyards, and more. We clarify this in the manuscript and we point to our schools and religious organizations findings as credible placebo tests.

3.12. Line 433. This doesn’t seem like a reasonable assumption. If anything, it seems like consumption preferences would differ between cell phone and no-cell phone families. If this is the case, you may actually be overstating the results, not understating them. Absent some supporting evidence, it might be wise to just cut the sentence.

Response: We agree with the reviewer’s comment. We now better qualify our statement in the Measurement Validation section and drop it from the Discussion section (where it could be confusing). If families with and without smartphones have similar preferences for consumption (of childcare as well as other items), this implies that our results likely *understate* the CTC’s impact on consumption, as we would be missing relatively more information for the group that is most likely to benefit from the policy change. If, instead, families without smartphones have different consumption preferences, for instance are averse to the use of childcare, then our estimates would *overstate* the effect of interest.

A few new comments

3.13. The organization of the paper is confusing. I did see that the “results-before-methodology” change is due to journal policies, but there were other changes that also make the manuscript harder to read. For example, the discussion should come after we are familiar with the methodology and data. It would be much better to mirror the first manuscript’s organization as close as possible while still checking whatever boxes are necessary at the journal.

Response: We appreciate the reviewer’s recommendation and also for acknowledging that we are constrained here by journal guidelines. We have taken the editor’s suggestion to add greater detail on data and identification strategy at the start of the results section, while leaving some of the denser methodological detail to the Methods section. We hope this provides sufficient information for the readers before getting into the results.

3.14. It would be more convincing if you control directly for county-level wages, even if they are imperfect (see the QCEW). Combined with the unemployment rate, this would go a long way to convincing readers that you are not attributing other economic factors to the CTC response.

Response: We now include the county-level wage information into our model controls. Specifically, we include the log of “average weekly wages” by counties across workers in all sectors during the quarter, which comes from the QCEW database that the reviewer referenced.

3.15. Line 192: I’m not sure most people considered the policy change temporary (many assumed it would be made permanent). But let’s say they did assume it was temporary: why would it reduce incentives to find stable work? Are you referring to people on the margin who couldn’t afford formal care without the more generous credit? This should be clarified or the sentence could be deleted.

Response: For what concerns perceptions about the nature of the policy change, data on expectations about the temporary vs permanent nature of the expanded CTC have been collected by the RAPID-EC Survey of the Stanford Center on Early Childhood in March 2021. The survey asks parents with at least one child age 5 or under nationwide whether they believe monthly CTC payments are temporary, potentially permanent, or permanent. In the early months of the CTC expansion, 65% of parents believe that the Child Tax Credit is temporary, 29% believe that it is potentially permanent, and 6% believe the Child Tax Credit is permanent. A higher percentage of parents in lower-income households (69%) believe that the Child Tax Credit is temporary

compared to those of middle-income (63%) and higher-income (62%) households. Lower-income is defined as pre-pandemic income 200% or more below federal poverty level (FPL); middle-income as between 200%–400% of FPL; Higher-income as 400% or more above FPL.

In addition, major institutional websites emphasized the temporary nature of the policy. For instance, the White House FAQs on the expanded CTC report: “**Q:** Will I keep getting the expanded credit amounts and the advance payments next year? **A:** The American Rescue Plan enacted these historic changes to the Child Tax Credit for 2021 only.” Similarly, the Internal Revenue Service CTC webpage reports: “There have been important changes to the Child Tax Credit that will help many families receive advance payments. The American Rescue Plan Act (ARPA) of 2021 expands the Child Tax Credit (CTC) for tax year 2021 only.”

Overall, these pieces of evidence seem to support the notion that the policy change was communicated and perceived as temporary. As the reviewer correctly points out, the policy being perceived as temporary would reduce the incentives to find stable work for individuals on the margin, who could not afford formal care – or other participation-related expenses – without the more generous credit. We have clarified this point in the manuscript.

3.16. Please include the event study of the school visit placebo in the Appendix. This is an important outcome that will be on the minds of a lot of people so you should emphasize it more. In fact, it would be wise to show how your results change when you omit vs include this as a control variable (line 386).

Response: We have now added the event study of the school visit placebo into Appendix E. We have also added in the placebo test for religious institutions there. We clarify in the Appendix that our mobility results are consistent (coefficients are slightly larger; thus, our preferred point estimates are more conservative) when not controlling for school visits.

3.17. The pre-trend checks are fine, even if they usually turn out to be null due to reduced sample sizes. Most of the action we are concerned about comes later in the year as schools and local economies adjusted to the later stages of COVID. What I’m getting at is distinguishing between pre-trends biasing your estimates and contemporaneous factors correlated with poverty bins.

Response: We agree and fully recognize this identification issue. We attempt at controlling for COVID-related factors that may be systematically associated with poverty bins in the post-treatment period, by including a set of controls aimed at capturing the impact of COVID on local economies. Our placebo tests failing to detect any association between school visits and poverty indicators for the CTC treatment periods is also reassuring of our ability to control for contemporaneous factors that would otherwise bias our results. We also replicate our core findings with state-by-month fixed effects, as discussed previously in this revision note and in the manuscript.

3.18. Upon closer inspection, I am confused as to the grouping of cars, clothes, hobbies, and home expenditures. These should be split up into more than one category. There are more than enough

observations to do so. Cars and home improvement supplies were greatly affected by supply chains while clothing and hobbies were not. There may be different propensities to spend the CTC on those subcategories, too. It isn't that surprising to find a null effect given this random grouping. If there is a better justification for grouping those categories together, please include it.

Response: We now include an Appendix that presents results for each of these categories individually. The results do not show meaningful differences in visits to the four categories. We now clarify in the manuscript that we group these four categories together to represent more-durable goods rather than services and less-durable goods that compose our other categories. To maintain some parsimony in the analyses, we maintain the combined categories in our primary analyses, but present the disaggregated categories in the Appendix. We thank the reviewer for this and all other suggestions for improving the paper.

Decision Letter, third revision:

20th October 2023

Dear Dr. Parolin,

Thank you for your patience as we've prepared the guidelines for final submission of your Nature Human Behaviour manuscript, "Consumption Responses to an Unconditional Child Allowance in the United States" (NATHUMBEHAV-23010152C). Please carefully follow the step-by-step instructions provided in the attached file, and add a response in each row of the table to indicate the changes that you have made. Please also address the additional marked-up edits we have proposed within the reporting summary. Ensuring that each point is addressed will help to ensure that your revised manuscript can be swiftly handed over to our production team.

We would hope to receive your revised paper, with all of the requested files and forms within two-three weeks. Please get in contact with us if you anticipate delays.

If you have not done so already, please alert us to any related manuscripts from your group that are under consideration or in press at other journals, or are being written up for submission to other journals (see:

<https://www.nature.com/nature-research/editorial-policies/plagiarism#policy-on-duplicate-publication> for details).

Nature Human Behaviour offers a Transparent Peer Review option for new original research manuscripts submitted after December 1st, 2019. As part of this initiative, we encourage our authors to support increased transparency into the peer review process by agreeing to have the reviewer comments, author rebuttal letters, and editorial decision letters published as a Supplementary item. When you submit your final files please clearly state in your cover letter whether or not you would like to participate in this initiative. Please note that failure to state your preference will result in delays in accepting your manuscript for publication.

In recognition of the time and expertise our reviewers provide to Nature Human Behaviour's editorial process, we would like to formally acknowledge their contribution to the external peer review of your manuscript entitled "Consumption Responses to an Unconditional Child Allowance in the United States". For those reviewers who give their assent, we will be publishing their names alongside the published article.

Cover suggestions

We welcome submissions of artwork for consideration for our cover. For more information, please see our https://www.nature.com/documents/Nature_covers_author_guide.pdf target="new"> guide for cover artwork.

ORCID

Non-corresponding authors do not have to link their ORCIDs but are encouraged to do so. Please note that it will not be possible to add/modify ORCIDs at proof. Thus, please let your co-authors know that if they wish to have their ORCID added to the paper they must follow the procedure described in the following link prior to acceptance:

Nature Human Behaviour has now transitioned to a unified Rights Collection system which will allow our Author Services team to quickly and easily collect the rights and permissions required to publish your work. Approximately 10 days after your paper is formally accepted, you will receive an email in providing you with a link to complete the grant of rights. If your paper is eligible for Open Access, our Author

Services team will also be in touch regarding any additional information that may be required to arrange payment for your article.

Please note that *Nature Human Behaviour* is a Transformative Journal (TJ). Authors may publish their research with us through the traditional subscription access route or make their paper immediately open access through payment of an article-processing charge (APC). Authors will not be required to make a final decision about access to their article until it has been accepted. Find out more about Transformative Journals

[REDACTED]

Best regards,
Alex McKay
Editorial Assistant
Nature Human Behaviour

On behalf of

Aisha Bradshaw, PhD
Senior Editor
Nature Human Behaviour

Reviewer #3:
None

Final Decision Letter:

Dear Dr. Parolin,

We are pleased to inform you that your Article "Consumption Responses to an Unconditional Child Allowance in the United States", has now been accepted for publication in *Nature Human Behaviour*.

Please note that *Nature Human Behaviour* is a Transformative Journal (TJ). Authors may publish their research with us through the traditional subscription access route or make their paper immediately open access through payment of an article-processing charge (APC). Authors will not be required to make a final decision about access to their article until it has been accepted. Find out more about Transformative Journals

With best regards,
Aisha

Aisha Bradshaw, PhD

Senior Editor
Nature Human Behaviour